# Psilocybin during the postpartum period induces long-lasting adverse effects in both mothers and offspring

Cassandra J. Hatzipantelis [1,2], Min Liu [1,3], Adam Love[4], Sadie J. Leventhal [4], Hero Maera[1,4], Srinidhi Viswanathan[4], Emily Avetisyan [4], Liana Belinsky[4], McKenna M. Rangel[4], Nina J. Jain [5], Max Kelly[6], Claire Copeland [4], Yara A. Khatib [1,7], Oliver Fiehn [1,3], David E. Olson [1,2,8,9] ✉ & Danielle S. Stolzenberg [1,4,10] ✉

Psilocybin increases social connectedness and has strong clinical transdiagnostic efficacy for mental illness, making it a candidate treatment to reduce maternal disconnect, anxiety, and blunted affect seen in peripartum mood disorders. However, the efficacy and safety of psilocybin in peripartum mood disorders has not been investigated. We used a social stress model to examine the effects of psilocybin in parous mice and their offspring. Social stress induced maternal withdrawal and increased stress-related behaviors – none of which were ameliorated by psilocybin. Weeks later, psilocybin-treated dams were more anxious, regardless of stress exposure. In contrast, psilocybin-treated virgin females were unaffected. Though reproductive status did not affect psilocybin pharmacokinetics, serotonin receptor transcription and 5-HT2A receptor-dependent responses were reduced in dams. Offspring exposed to maternal psilocybin during breastfeeding exhibited anhedonia in adulthood. Here, we show that both parous parents and their children may be uniquely vulnerable to psychedelic treatment during the postpartum period.

Peripartum mental illness is responsible for 1 in 4 maternal deaths in the United States[1]. Mood symptoms are so common following birth, the term baby blues has been used to describe the anxiety, irritation, and depression that occurs following 80% of births[2]. More than 20% of birthing parents experience prolonged mood symptoms during pregnancy or the postpartum period, leading to a peripartum mood disorder (PMD) diagnosis[3–6]. Anxiety and obsessive-compulsive symptoms commonly co-occur with depressive symptoms, and in the postpartum period, birthing parents report the most distress over difficulty bonding with their infants[7,8]. Although the DSM-V peripartum

onset specifier is for mood episodes that begin during pregnancy or within 4 weeks of delivery[9], at least half of birthing parents who report no mood symptoms in the first 2–6 months postpartum end up with a diagnosis after 6 months[10] and 60% of maternal suicides occur outside of the immediate postpartum period (43–365 days postpartum)[11]. Postpartum physiology can confer a longer-term vulnerability to mood disorders with mood symptoms lasting up to 11 years after birth[12]. To be effective, treatments for PMDs must be fast-acting, accessible, and safe for developing offspring who may become exposed. Rapid antidepressant and anxiolytic effects are essential as early mother–infant

[1]Institute for Psychedelics and Neurotherapeutics, University of California, Davis, Davis, USACA. [2]Department of Chemistry, University of California, Davis, Davis, CA, USA. [3]West Coast Metabolomics Center, University of California, Davis, Davis, CA, USA. [4]Department of Psychology, University of California, Davis, Davis, CA, USA. [5]Global Disease Biology, University of California, Davis, Davis, CA, USA. [6]Genetics and Genomics, University of California, Davis, Davis, CA, USA. [7]Pharmacology and Toxicology Graduate Group, University of California, Davis, Davis, CA, USA. [8]Department of Biochemistry & Molecular Medicine, University of California, Davis, Sacramento, CA, USA. [9]Center for Neuroscience, University of California, Davis, Davis, CA, USA. [10]Perinatal Origins of Disparities Center, University of California, Davis, Sacramento, CA, USA. ✉e-mail: deolson@ucdavis.edu; dstolzenberg@ucdavis.edu

interactions are critical for bonding and the maternal avoidance or withdrawal that often accompanies PMDs can have long-term effects on attachment[13].

Psychedelics like psilocybin are being explored as anti-depressants given that they can rapidly reduce mood-related symptoms by 60–80% after a single dose with effects lasting for up to 12 weeks[14]. In addition to antidepressant effects, patients taking psilocybin have described increased connectedness with loved ones and self-compassion, underscoring the potential utility of these fast-acting, long-lasting single-dose treatments for addressing the debilitating parent–infant disconnect and personal shame symptoms associated with PMDs. Clinical trials are ongoing to determine if a novel psilocybin analog could address peripartum mood disorders [NCT06342310]. Thus, there is an urgent need to understand if psychedelics administered during the postpartum period are safe and effective.

Our group has recently developed a social stress paradigm to model postpartum mood symptoms in C57BL6/J (B6) mice[15]. We have used a 2-cage system that enables quantification of active maternal avoidance, which is a hallmark of postpartum depression. Repeated exposure to a social threat experience destabilizes maternal behavior, induces pup avoidance and triggers prolonged stress-related behaviors. These behavioral changes remain constant 24 h later, even when no social threat is present. In the present study, we used this paradigm to investigate the effects of a single psilocybin dose on postpartum maternal care as well as anxiety- and depressive-like behaviors after offspring were weaned. We also examined social, emotional and cognitive behavioral outcomes in adult mice who were reared by dams exposed to maternal stress and/or psilocybin treatment. This design allowed us to disentangle the effects of both maternal stress and psilocybin treatment as well as investigate their combined impact on parous mice and their offspring.

Here we show that psilocybin does not ameliorate postpartum social stress but rather increases anxious behavior weeks later regardless of stress exposure. Reproductive status likely mediates the adverse effects of psilocybin postpartum, as virgin females were unaffected. Though psilocybin pharmacokinetics do not vary by reproductive status, dams show less serotonin receptor transcription and reduced 5-HT2A receptor-dependent responses. Finally, male and female offspring reared by psilocybin-exposed dams exhibit anhedonia in adulthood, an effect that is recapitulated by directly exposing pups to psilocin during development. These data suggest that the postpartum period may represent a unique period of vulnerability to psilocybin treatment, and psychedelic use may pose a risk for both newly parous parents and their children.

## Results

### Social stress impairs maternal behavior
Dams subjected to three days of social stress (Fig. 1A) displayed significantly impaired maternal behaviors. On average, stressed dams took significantly longer than control mice to respond to pups upon reunion (significant increase in pup retrieval latency; $p = 0.0138$; Fig. 1B) across the 3 intruder test days. Stressed mice were also more likely than control dams to circle the cage with pups in their mouth and carry pups to locations outside of the nest area (59% versus 18%, $p = 0.0078$; Fig. 1C), and by intruder day three, 56% of stressed dams had relocated their nests compared to 14% of control dams (stress effect; $p = 0.0031$; Fig. 1D). Once in the nest, and irrespective of intruder day, stressed dams spent significantly less time in contact with the full litter (main stress effect; $F(1,47) = 19.42$, $p < 0.0001$; Fig. 1E), specifically less time nursing their pups (main stress effect; $F(1,47) = 6.98$, $p = 0.011$; Fig. 1F), and less time sniffing/licking their pups (main stress effect; $F(1,47) = 16.75$, $p = 0.0002$; Fig. 1G). Stressed dams spent more time exhibiting stress-related behaviors such as interacting with some but not all pups, preventing pups from nursing,

or engaging in self-grooming, digging, or immobility outside of the nest (main stress effect; $F(1,47) = 51.19$, $p < 0.0001$, Fig. 1H). Stressed mice also spent more time actively avoiding their litters in an entirely different cage (main stress effect; $F(1,47) = 19.26$, $p < 0.0001$; Fig. 1I). Finally, social stress reduced the consistency of care provided to pups by dams (main stress effect; $F(1,47) = 19.53$, $p < 0.0001$; Fig. 1J). Importantly, all above observations were in accordance with previously published results using this model[15].

### Psilocybin does not treat maternal stress and induces stress itself
To assess the effects of a single dose of psilocybin on the sustained impact of social stress, psilocybin was administered ~1 h following the final maternal care assessment after the third intruder exposure on postpartum day (PPD) 7 and maternal care was assessed 24 h later, on PPD8 (Fig. 2A). There were no sustained effects of social stress nor psilocybin treatment on pup retrieval measures ($p > 0.05$; Fig. 2B), whereas maternal care behaviors were significantly impacted by either stress or psilocybin. Social stress-exposed dams showed a sustained reduction in contact with the full litter (main stress effect; $F(1,45) = 5.290$, $p = 0.0261$; Fig. 2C), including a reduction in nursing (main stress effect; $F(1,45) = 4.735$, $p = 0.0348$; Fig. 2D), which was not affected by psilocybin treatment. While there was no sustained effect of intruder stress on sniffing/licking of pups, psilocybin treatment significantly reduced this maternal behavior regardless of stress exposure (main psilocybin effect; $F(1,45) = 5.003$, $p = 0.0303$; Fig. 2E). Both stress and psilocybin increased stress-related behaviors (main stress effect; $F(1,42) = 6.345$, $p = 0.0157$; main psilocybin effect; $F(1,42) = 4.697$, $p = 0.0359$; Fig. 2F). Again, there was a sustained impact of stress on pup avoidance (main stress effect; $F(1,43) = 11.99$, $p = 0.0012$; Fig. 2G. Finally, no sustained effect of stress was observed for fragmented care ($p > 0.05$; Fig. 2H).

Using an integrative technique of combining z-normalized results across independent maternal care behavior measurements, both stress and psilocybin exposure clearly increase the risk of impairments in maternal care even 24 h later (main stress effect; $F(1,39) = 12.57$, $p = 0.0010$; main psilocybin effect; $F(1,39) = 5.487$, $p = 0.0244$; Fig. 2I). However, there was no significant interaction between stress and psilocybin ($p = 0.6959$), suggesting that the effects of stress and psilocybin were not synergistic, but distinct. Notably, the effect size of social stress' contribution to sustained maternal care impairments ($\eta^2 = 0.20$, large) was greater than that of psilocybin ($\eta^2 = 0.09$, medium). While neither social stress nor psilocybin influenced the sucrose preference of dams, the sucrose preference of saline-treated stressed dams negatively correlated with pup avoidance ($p = 0.0005$; Supplementary Fig. 2). This correlation was not observed for psilocybin-treated stress dams ($p = 0.1041$).

### Postpartum psilocybin negatively affects mood-related phenotypes
Two weeks following injection (Fig. 3A), psilocybin-treated mice possessed an anxiogenic phenotype, spending significantly less time in the center zone of the open field test (main psilocybin effect; $F(1,38) = 14.36$, $p = 0.0005$; Fig. 3B) and significantly less time in the open arms of the elevated plus maze (main psilocybin effect; $F(1,44) = 5.406$, $p = 0.0247$; Fig. 3C). Exposure to stress or psilocybin 2 weeks prior had no influence on cognition as measured in the novel object recognition task, nor on affect as measured in the splash test and forced swim test ($p > 0.05$; Fig. 3D–F).

Combining z-normalized results across all tests, psilocybin exposure clearly increases the risk of behavioral impairments in this battery two weeks later (main psilocybin effect; $F(1,45) = 7.925$, $p = 0.0072$; Fig. 3G). Notably, the effect size of psilocybin-induced increases in long-term behavioral risk ($\eta^2 = 0.144$, large) was greater than that of the social stress ($\eta^2 = 0.023$, small). These effects are

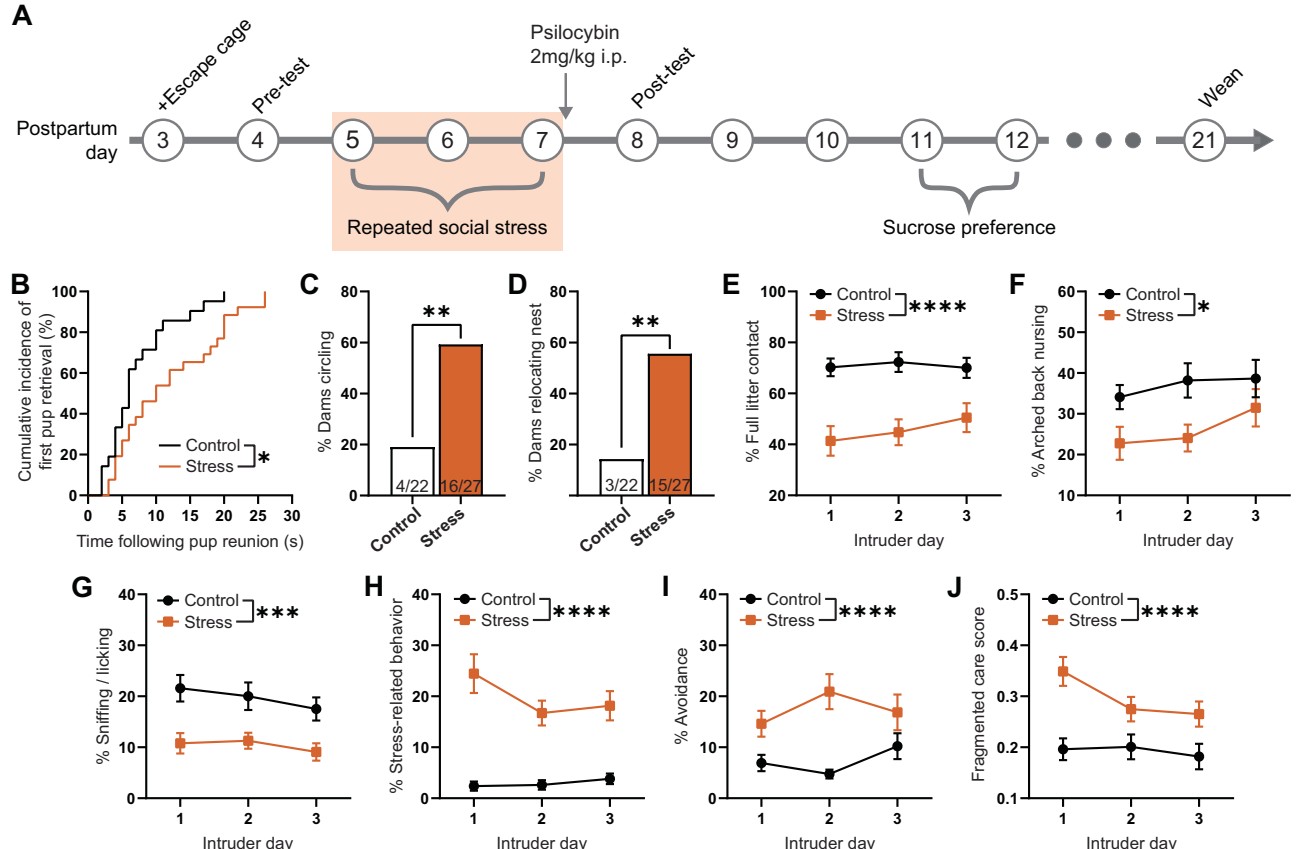

**Fig. 1 | Social stress impairs pup retrieval and maternal care following pup reunion. A** Timeline of experiment. Data illustrated in **B**–**J** represent maternal behaviors exhibited by control ($n = 22$) and stressed ($n = 27$) dams, prior to drug administration, measured during the 1 h observation period following intruder exposure on postpartum days 5–7 (shaded in **A**). **B** Percentage of dams retrieving their first pup to the nest following pup reunion. **C** Percentage of dams that displayed circling behavior on any testing day. **D** The percentage of dams relocating their nest following at least one intruder exposure. Percentage of observations made (**E**) in contact with the entire litter, **F** engaging in arched back nursing, **G** sniffing or licking their pups, **H** exhibiting stress-related behaviors, **I** avoiding litter in the cage not containing the nest. **J** Fragmented care score indicating behavioral switching between pup-directed and non-pup-directed behaviors. Data in **B** are presented as a survival plot and comparisons between control and stress groups are made using a Log-rank Mantel-Cox test. Proportion data in **C**, **D** are presented as bar graphs representing results from contingency analyses with statistical comparisons between groups made using Chi-square (and Fisher's exact) test (two-sided). Data in **E**–**J** are presented as mean ± SEM and comparisons between control and stress groups made using a two-way analysis of variance. $*p < 0.05$, $**p < 0.01$, $***p < 0.001$ and $****p < 0.0001$. Exact *P* values are provided in the Source Data.

specific to the postpartum period as virgin female mice treated with a single dose of psilocybin had no, if not decreased, risk of behavioral impairments in the battery using the z-normalized results ($p > 0.05$; $d = 0.86$, large; Fig. S3), and in contrast to parous females, produced an anxiolytic effect in the open field test ($p = 0.0451$; Supplementary Fig. 3).

**Maternal psilocybin has long-term adverse effects on offspring**

Adult offspring of experimental dams were subjected to a battery of behavioral tests (Fig. 4A). Adult offspring reared by socially stressed mothers displayed an anxiogenic phenotype in the open field test (main stress effect; $F(1,68) = 4.387$; $p = 0.0399$; Fig. 4B). Meanwhile there were no long-term effects on working memory in the T-maze spontaneous alternation task ($p > 0.05$; Fig. 4C). While neither maternal stress nor psilocybin exposure affected sociability of offspring ($p > 0.05$; Fig. 4D), male offspring reared by socially stressed mothers showed impaired preference for social novelty, opting to spend less time in the chamber with a novel mouse over the chamber with their cage mate than other experiment groups (main stress effect in males; $F(1,38) = 6.014$; $p = 0.0189$; Fig. 4E). Offspring of psilocybin-treated mothers displayed more anhedonia in the sucrose preference test than offspring of saline-treated mothers (main psilocybin effect; $F(1,74) = 7.399$; $p = 0.0081$; Fig. 4F), although there

were no differences between groups in the forced swim test ($p > 0.05$; Fig. 4G).

Comparing z-normalized results across all tests, offspring reared by socially stressed mothers were found to be at greater behavioral risk than offspring reared by controls (main stress effect; $F(1,77) = 4.147$; $p = 0.0451$; $\eta^2 = 0.046$, medium; Fig. 4H) and maternal psilocybin exposure increased the risk of behavioral impairments in offspring regardless of their sex or maternal stress-exposure (main psilocybin effect; $F(1,77) = 4.677$; $p = 0.0337$; $\eta^2 = 0.052$, medium). There was also a small effect of sex on offspring behavioral risk ($\eta^2 = 0.026$, small).

**Developmental psilocin exposure explains offspring anhedonia**

We designed a pharmacokinetics experiment allowing us to simultaneously assess the acute effects of psilocybin exposure on maternal care behaviors whilst also measuring the pharmacokinetics of psilocybin and psilocin in the brains of breast-feeding pups (Fig. 5A). We collected dam brains at the conclusion of the experiment 2 h following psilocybin exposure, alongside the brains of virgin females also exposed to psilocybin 2 h prior. We found no differences in psilocin levels (Fig. 5B) in the brains of postpartum or virgin females 2 h following psilocybin injection ($p > 0.05$), however, at least some psilocin was transferred to the breastmilk as psilocin was detected in the brains of nursing pups (Fig. 5C). Specifically, peak detection

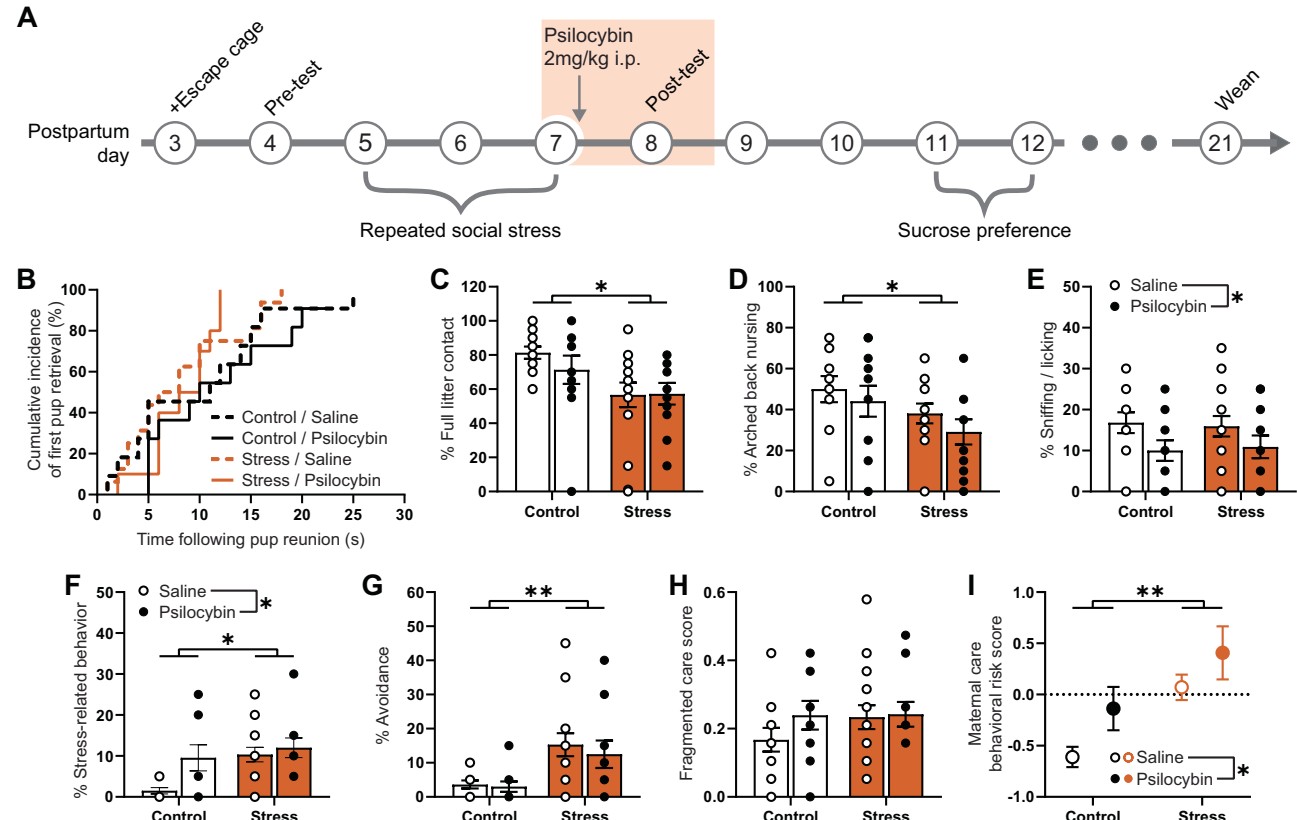

**Fig. 2 | Psilocybin does not treat social stress-induced impairments in maternal care. A** Timeline of experiment. Data illustrated in **B**–**H** represent maternal behaviors exhibited by control dams exposed to saline ($n = 11$) or psilocybin ($n = 11$), and stressed dams exposed to saline ($n = 16$) or psilocybin ($n = 11$) measured during a 1 h observation period 24 h following final intruder and psilocybin exposure on PPD 8 (shaded in **A**). Animals used in these experiments are the same as those used in Fig. 1. **B** Cumulative percentage of dams having retrieved their first pup to the nest following pup reunion. Percentage of observations made (**C**) in contact with the entire litter, (**D**) engaging in arched back nursing, (**E**) sniffing or licking their pups, (**F**) exhibiting stress-related behaviors, (**G**) avoiding litter in the cage not containing the nest. **H** Fragmented care score indicating behavioral switching between pup-directed and non-pup-directed behaviors. **I** Integrative maternal care behavioral risk score, an integrative measure of global maternal care impairments calculated by averaging z-normalized results from independent maternal care behaviors (arched back nursing, sniffing/licking, stress-related behaviors, avoidance and fragmented care), such that the greater the risk score, the greater the global impairment. Data in **B** are presented as a survival plot and comparisons between groups are made using a Log-rank Mantel–Cox test. Individual data in **C**–**H** are presented as aligned dot plots and as bar charts representing mean ± SEM. Given that there are a discrete number of observations able to be made in the 1 h observation period, the results from multiple dams may be represented by a single point on the graph. Data in **I** are presented as mean ± SEM. Statistical comparisons between experimental groups for data in **C**–**I** are made using a two-way analysis of variance. *$p < 0.05$, **$p < 0.01$. Exact $P$ values are provided in the Source Data.

occurred at 30 min following reunion of mom and pups where 60% of the samples showed detectable levels of psilocin, compared to 80% by 1 h and 100% by 2 h. During that time, psilocybin dramatically disrupted maternal care behaviors when compared to baseline performance measured the day prior (Supplementary Fig. 4), with a very large ($d = 2.97$) increase in the maternal care behavioral risk score ($p = 0.0030$, Supplementary Fig. 4F). Untransformed psilocybin was not detected in any sample.

To determine whether this direct exposure to psilocin could explain the behavioral deficits in offspring of dams exposed to psilocybin (Fig. 4), psilocin was administered directly to pups on PPD 7, before being reared to adulthood and subjected to the same battery of behavioral tests as the previous experiment (Fig. 5D). Direct exposure of pups to psilocin had no effect on anxiety-like behavior in the open field test, spontaneous alternations in the T maze, sociability, or preference for social novelty in 3-chamber social interaction tests ($p > 0.05$; Fig. 5E–H). Meanwhile, both male and female pups exposed to psilocin displayed more anhedonia in the sucrose preference test than saline-exposed littermates (main psilocin effect; $F(1,45) = 7.203$; $p = 0.0102$; Fig. 5I), although there were no differences between groups in the forced swim test ($p > 0.05$; Fig. 5J).

While these findings in specific behaviors perfectly replicate the phenotype of offspring of dams exposed to psilocybin, there was only a small, non-significant effect of psilocin when comparing overall behavioral risk ($p > 0.05$; $\eta^2 = 0.008$, small; Fig. 5K). There was also a moderate effect of sex whereby females were found to be at greater behavioral risk in the battery than males ($p > 0.05$; $\eta^2 = 0.048$, medium; Fig. 4H). While there was no significant interaction effect ($p > 0.05$), the effect of psilocin exposure on females appeared to be greater in females ($d = 0.33$, small-medium) than in males ($d = 0.04$), much like observed previously.

### Serotonergic agonists and antagonists acutely impair maternal care

To assess whether psilocybin-induced effects on maternal care behaviors were dependent on activity at serotonin (5-HT) receptors, dams and yoked virgin female mice were pre-treated with the 5-HT2 receptor antagonist, ketanserin, 10 min prior to psilocybin administration. Pharmacodynamics was then assessed by head-twitch response immediately prior to 1 h of maternal care assessments (Fig. 6A). Head-twitch analysis revealed a significant three-way interaction (psilocybin x ketanserin x reproductive status interaction effect; $F(1,79) = 75.21$,

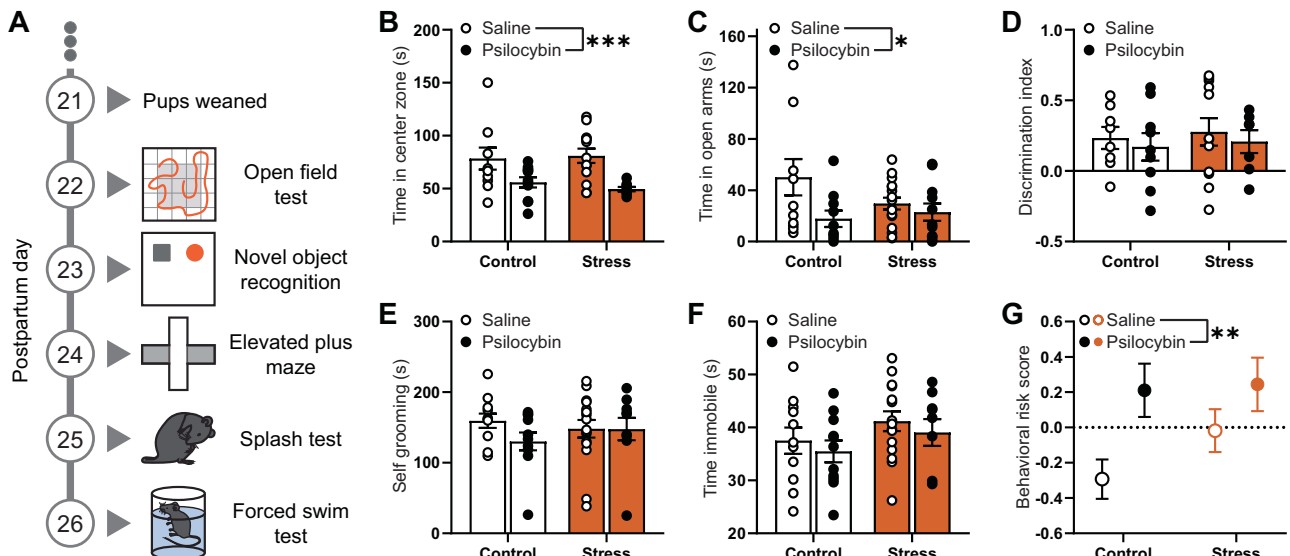

**Fig. 3 | Psilocybin produces long-term mood disorder-related phenotypes in parous mice. A** Timeline of behavioral battery. Data illustrated in **B**–**F** represent primary endpoints measured in the battery completed two weeks following final intruder and psilocybin exposure beginning on PPD 22 for saline ($n = 11$) or psilocybin treated control dams ($n = 11$), and saline ($n = 16$) or psilocybin treated stress dams ($n = 11$). **B** Time spent in the center zone of the open field test. **C** Time spent in the open arms of the elevated plus maze. **D** Discrimination index indicating the relative time spent investigating a novel object compared to a known object, ($t_{novel\ object} - t_{familiar\ object}$) / ($t_{novel\ object} + t_{familiar\ object}$). **E** Time spent self-grooming in the splash test. **F** Time spent immobile in the forced swim test. **G** Integrative behavioral risk score, an integrative measure of global impairments in the test battery calculated by averaging z-normalized results from the primary endpoints of each behavioral test for each animal, such that the greater the risk score, the greater the global impairment. Individual data in **B**–**F** are presented as aligned dot plots and as bar charts representing mean ± SEM. Data in **G** are presented as mean ± SEM. Statistical comparisons between experimental groups are made using a two-way analysis of variance. *$p < 0.05$, **$p < 0.01$, ***$p < 0.001$. Exact $P$ values are provided in the Source Data.

$p < 0.0001$; Fig. 6B) whereby psilocybin increased headtwitches to a greater extent in virgin females compared to postpartum females (psilocybin x reproductive status interaction effect; $F(1,79) = 75.21$, $p < 0.0001$) although both are completely abolished by ketanserin pre-treatment.

To assess whether this decreased head-twitch response in the postpartum period compared to the virgin state was due to changes in 5-HT receptor levels, we used real-time qPCR to assess relative differences in the expression of 5-HT receptors that are highly responsive to psilocin in the cortex of postpartum females, virgin females, and virgin male mice (Fig. S5). We found that reproductive state/sex significantly altered the expression of mRNA encoding multiple 5-HT receptors (5-HT1A, 2A and 2C) as well as *Bdnf* regardless of transcript (main effect; $F(2,59) = 8.545$, $p = 0.0006$). Planned comparisons highlighted that postpartum female mice displayed downregulated mRNA expression compared to both virgin females ($p = 0.0004$) and virgin males ($p = 0.0347$).

Psilocybin significantly increased first pup retrieval latency ($p = 0.0438$; Fig. 6C), while ketanserin had no effect ($p > 0.05$). Drug treatment also significantly impacted latency to nurse ($p < 0.0001$; Fig. 6D), with psilocybin increasing nursing latency both alone ($p = 0.0006$) and in combination with ketanserin ($p = 0.0021$), although ketanserin alone did not affect latency to nurse ($p > 0.05$).

Psilocybin and ketanserin independently produced acute reductions in contact with the full litter (main psilocybin effect; $F(1,42) = 7.143$, $p = 0.0107$; main ketanserin effect; $F(1,42) = 7.403$, $p = 0.0094$; Fig. 6E), although only ketanserin significantly reduced nursing (main ketanserin effect; $F(1,42) = 7.270$, $p = 0.0100$; Fig. 6F) while only psilocybin significantly reduced sniffing/licking of pups (main psilocybin effect; $F(1,42) = 5.885$, $p = 0.0196$; Fig. 6G). Both psilocybin and ketanserin independently showed acute increases in stress-related behaviors (main psilocybin effect; $F(1,42) = 8.104$, $p = 0.0067$; main ketanserin effect; $F(1,42) = 12.14$, $p = 0.0012$; Fig. 6H),

while neither impacted fragmented care compared to saline ($p > 0.05$; Fig. 6I).

Using an integrative technique of combining z-normalized results across independent maternal care behavior measurements, both acute psilocybin and ketanserin exposure increase the risk of impairments in maternal care (main psilocybin effect; $F(1,42) = 10.36$, $p = 0.0025$; main ketanserin effect; $F(1,42) = 5.739$, $p = 0.0211$; Fig. 2I). However, there was no significant interaction between psilocybin and ketanserin ($p = 0.9886$), suggesting that the effects of psilocybin and ketanserin were not synergistic, but distinct. Notably, the effect size of psilocybin's contribution to acute maternal care impairments ($\eta^2 = 0.18$, large) was greater than that of ketanserin ($\eta^2 = 0.10$, medium).

## Discussion

The subjective experiences of increased connectedness with loved ones, self-compassion, and long-lasting remission of depressive symptoms often ascribed to psilocybin therapy align well with the needs of patients with PMDs. Moreover, the relatively rapid clearance of psychedelics from the body could allow for a quick return to breastfeeding, a behavior known to improve mental health outcomes for this patient population[16]. While clinical trials testing a psilocybin analog as a treatment for postpartum depression [NCT06342310] are currently ongoing, the phase 1 dose-escalation safety trial participants were predominantly male (~75%)[17]. Thus far, there has been no clinical or preclinical safety testing of psilocybin in the postpartum period, nor has the efficacy of psilocybin been assessed specifically in a preclinical model of postpartum mood disorders. Here, we describe the effects of a single dose of psilocybin to female mice in the postpartum period and detail the lasting adverse behavioral consequences for both mothers and their offspring.

Our model examined postpartum social stress and limited resources as factors that interfere with mother–infant bonding in the early postpartum period. Consistent with our previous work, we found

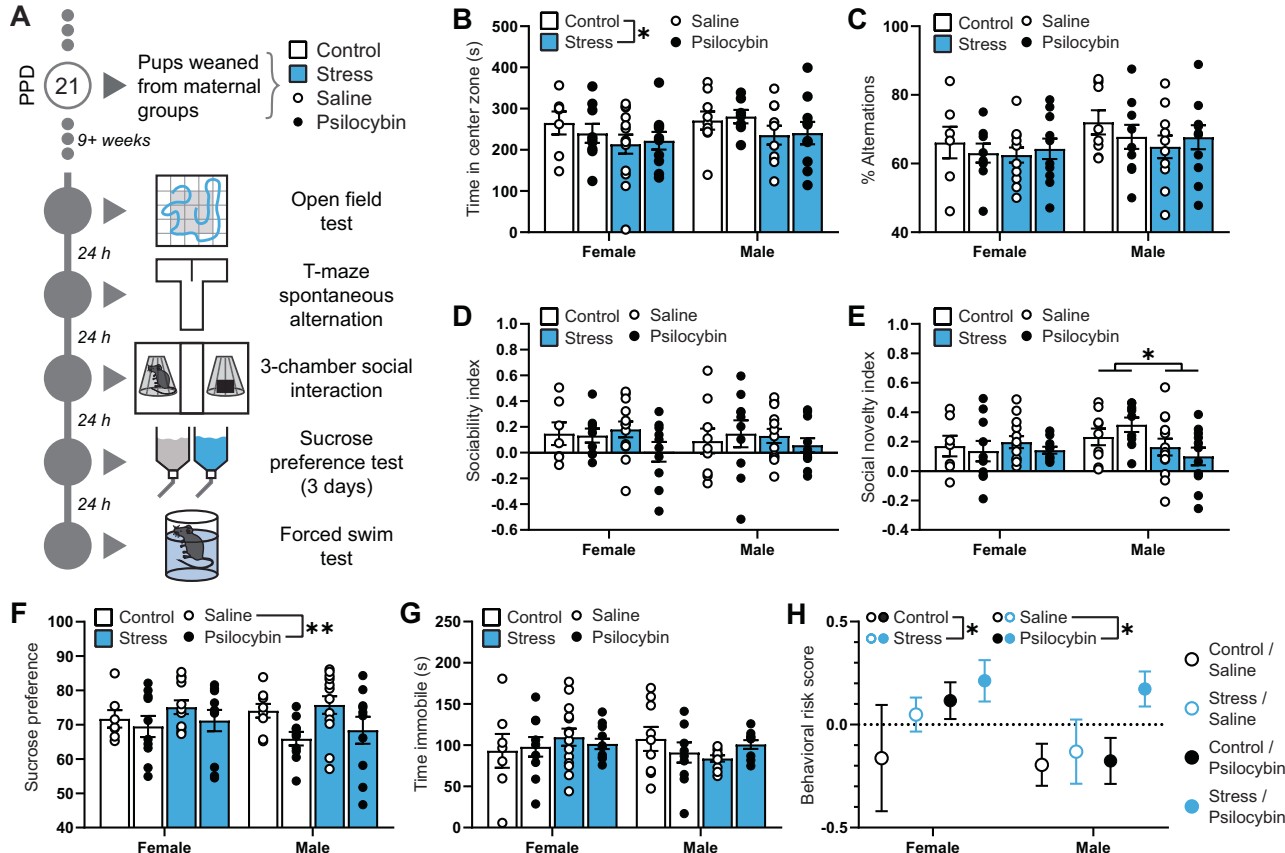

**Fig. 4 | Single postpartum exposure to psilocybin increases risk of behavioral phenotypes related to mood disorders in adult mice. A** Experimental timeline of offspring including time of wean from dams and sequence of behavioral battery. Data illustrated in **B–G** represent primary endpoints measured in the battery completed by adult female mice weaned from saline ($n = 7$) or psilocybin treated control dams ($n = 10$), adult male mice weaned from saline ($n = 9$) or psilocybin treated control dams ($n = 10$), adult female mice weaned from saline ($n = 14$) or psilocybin treated stress dams ($n = 11$), and adult male mice weaned from saline ($n = 13$) or psilocybin treated stress dams ($n = 11$). **B** Time spent in the center zone of the open field test. **C** Percentage of time spent making alternations in the T-maze spontaneous alternation task. **D** Sociability index indicating the relative time in the interaction chamber of a novel object compared to a familiar mouse, ($t_{mouse}$ − $t_{object}$) / ($t_{mouse}$ + $t_{object}$). **E** Social novelty index indicating the relative time spent in

the interaction chamber of a novel mouse compared to a familiar mouse, ($t_{novel\ mouse}$ − $t_{familiar\ mouse}$) / ($t_{novel\ mouse}$ + $t_{familiar\ mouse}$). **F** Sucrose preference score indicating the ratio of 1% sucrose solution consumed in a 24 h period when given the option between that and water. **G** Time spent immobile in the forced swim test. **H** Integrative behavioral risk score, an integrative measure of global impairments in the test battery calculated by averaging z-normalized results from the primary endpoints of each behavioral test for each animal, such that the greater the risk score, the greater the global impairment. Individual data in **B–G** are presented as aligned dot plots and as bar charts representing mean ± SEM. Data in **H** are presented as mean ± SEM. Statistical comparisons between maternal experimental groups and sexes are made using a three-way analysis of variance. *$p < 0.05$, **$p < 0.01$. Exact *P* values are provided in the Source Data.

that this paradigm reduced several aspects of care including pup retrieval, nursing, grooming, and tactile contact with the litter[15]. Social stress also dysregulated care, resulting in frequent behavioral switching and maternal stress behaviors including treating pups like irrelevant, inanimate objects (Supplementary Movie 1). Finally, stressed dams spent more time actively avoiding their litters by moving to a separate cage. Many of these behaviors persisted 24 h later, even when no social stress was present[15]. While antidepressant effects of psilocybin typically emerge 24 h after treatment of non-parous animals, we found no antidepressant-like effects of psilocybin on stressed or control postpartum mice when assessed 24 h after treatment on PPD 8.

Not only was psilocybin unable to reverse stress-induced maternal care impairments in the postpartum period, but early postpartum exposure to psilocybin conferred long-lasting deficits in a battery of PMD-related behaviors assessing mood and cognition. Specifically, psilocybin-exposed postpartum mice exhibited an anxiogenic phenotype and a more nuanced change in cognitive and depressive-like behaviors, culminating in a dramatically increased behavioral risk score. This was in stark contrast to the effects of psilocybin in virgin females. Although very few preclinical studies of psychedelics have

included nonparous females[18], when tested, psilocybin has been found to produce acute and lasting anxiolytic and antidepressant effects in both virgin males and females[19–21]. We too found, even in the absence of stress, a single dose of psilocybin in virgin, nonparous female mice produced an anxiolytic phenotype that aligns with other preclinical and clinical data. Furthermore, psilocybin exposure had no effect on long term behavioral risk as measured by the same behavioral battery used to assess the postpartum mice. The fact that we observed a therapeutic effect in nonparous females supports the translational relevance of our findings and suggests that psilocybin may possess a distinct risk profile when administered during the postpartum period.

It is well known that pregnancy and the postpartum period represent stages of great physiological change to both the body and brain[22,23]. Given that ovarian hormones directly regulate serotonergic systems[24], the dramatic rise and fall of estrogens and progestins substantially alters the physiological landscape, including the pharmacokinetics and/or pharmacodynamics of drugs that impact serotonin signaling. The activity of enzymes known to be involved in the metabolism of psilocybin, such as monoamine oxidase-A and cytochrome P450s (CYPs), are dramatically upregulated during the postpartum

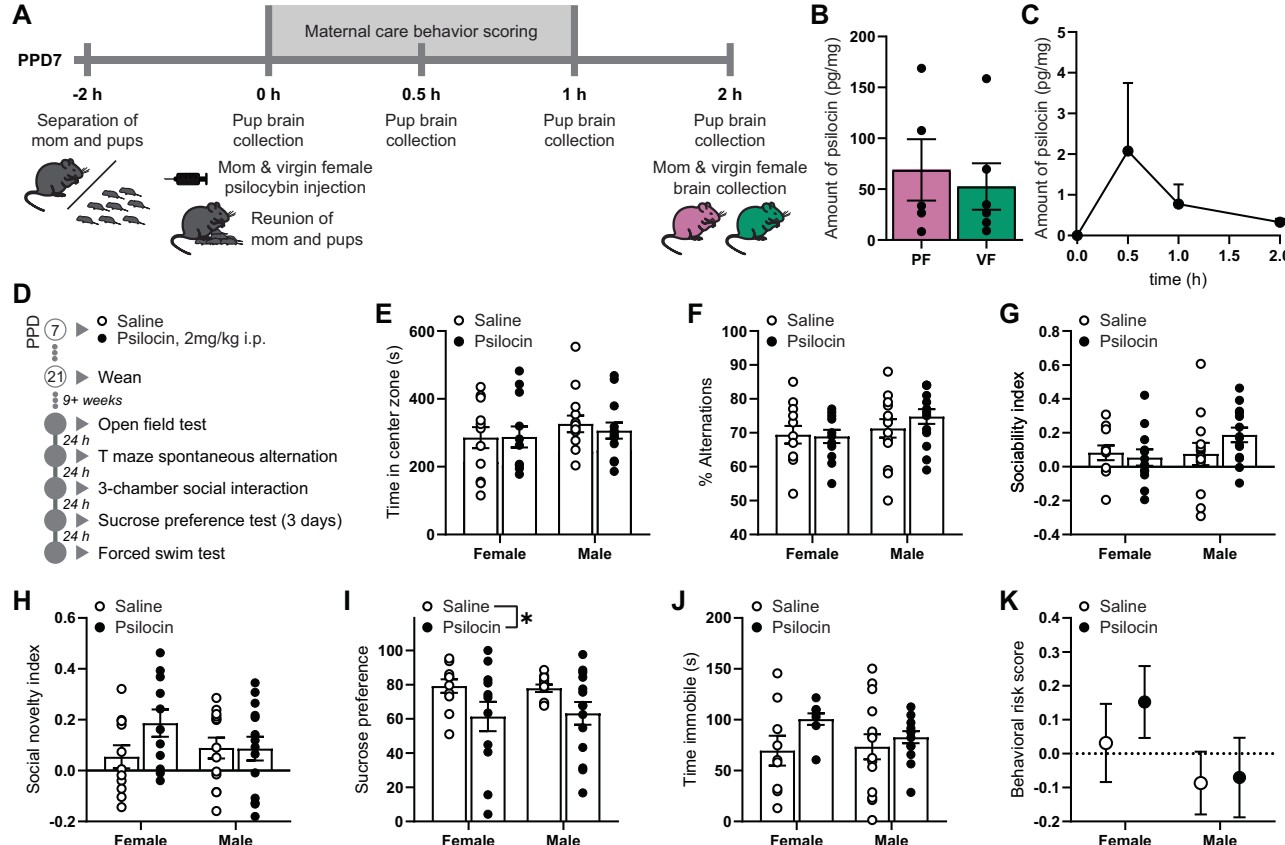

**Fig. 5 | Single exposure to psilocin during early postnatal development produces anhedonia in adulthood. A** Timeline of pharmacokinetics experiment allowing for the assessment of acute maternal care behaviors in response to psilocybin as well as the psilocybin pharmacokinetics of pups, dams and virgin females. **B** The pharmacokinetic time-course of psilocybin and psilocin levels in the brains of pups before maternal psilocybin injection ($n = 3$), or 30 min ($n = 5$), 1 h ($n = 5$), or 2 h ($n = 5$) following reunion. **C** The levels of psilocin present in the brains of postpartum females (PF; $n = 5$) and virgin females (VF; $n = 6$) 2 h following psilocybin treatment. **D** Timeline of behavioral battery. Data illustrated in **E–J** represent primary endpoints measured in the battery for adult female and male littermates treated with either saline ($n = 12,14$) or psilocin ($n = 12,14$) on postnatal day 7. **E** Time spent in the center zone of the open field test. **F** Percentage of time spent making alternations in the T-maze spontaneous alternation task. **G** Sociability index indicating the relative time in the interaction chamber of a novel object

compared to a familiar mouse, ($t_{mouse} - t_{object}$) / ($t_{mouse} + t_{object}$). **H** Social novelty index indicating the relative time spent in the interaction chamber of a novel mouse compared to a familiar mouse, ($t_{novel\ mouse} - t_{familiar\ mouse}$) / ($t_{novel\ mouse} + t_{familiar\ mouse}$). **I** Sucrose preference score indicating the ratio of 1% sucrose solution consumed in a 24 h period when given the option between that and water. **J** Time spent immobile in the forced swim test. **K** Integrative behavioral risk score, an integrative measure of global impairments in the test battery calculated by averaging z-normalized results from the primary endpoints of each behavioral test for each animal, such that the greater the risk score, the greater the global impairment. Individual data in **B**, **E–J** are presented as aligned dot plots and bar charts representing mean ± SEM. Data in **C**, **K** are presented as mean ± SEM and statistical comparisons in psilocin content are made using a $t$ test (two-sided). Statistical comparisons between psilocin- and saline-treated pups are made using a two-way analysis of variance. *$p < 0.05$. Exact $P$ values are provided in the Source Data.

period[25,26]. We didn't observe differences in psilocin levels in the brains of postpartum females compared to that of virgin females measured 2 h following psilocybin dosing. This may be because the 2 h time point is approaching the end of psilocin pharmacokinetics time course[27], and therefore differences in pharmacokinetics between parous and non-parous females may have been more easily detected if we established the full time course. However, the potency of psilocybin for inducing a head-twitch response is lower in the postpartum period. While psilocybin might be more rapidly metabolized in postpartum mice, any putative changes in pharmacokinetics is unlikely to explain the sustained adverse effects observed in dams and their offspring following postpartum administration of psilocybin. Future studies establishing the dose-response relationship of the efficacy and adverse event potential of psilocybin in both parous and non-parous female mice would help elucidate if any differences were as a result of changes in potency.

Beyond changes in drug metabolism, it is possible that psilocybin has a different effect on the brain of postpartum females because of differences in fundamental brain circuit functioning. The postpartum

period is associated with dramatic changes in neuroplasticity and adaptive restructuring, particularly in the early postpartum period (first 6 weeks for humans)[22,28]. Psilocin, the active metabolite of psilocybin, is known to promote structural and functional plasticity in cortical neurons through activation of 5-HT2ARs and the TrkB pathway[19,29–31]. Interestingly, we demonstrated that the expression of mRNA encoding 5-HT2A/2 C receptors as well as 5-HT1A receptors and the TrkB ligand BDNF were significantly downregulated in the cortex of postpartum females relative to virgin mice on postpartum day 7. Others have shown that these same 5-HT receptors are dynamically altered across reproduction and throughout parental neural circuits[32]. Our data are also consistent with reports that systemic administration of 5-HT2A or 5-HT2C receptor agonists disrupt maternal behavior and reduce pup preference in rats through executive control mechanisms or by impacting maternal motivation, respectively[33–36]. In the present study, we found that the 5-HT2 antagonist ketanserin, in the presence or absence of psilocybin, dramatically disrupted maternal care behaviors. Similarly, systemic 5-HT2A antagonist pre-treatment was unable to rescue 5-HT2A agonist-induced maternal deficits in rats and 5-HT2C

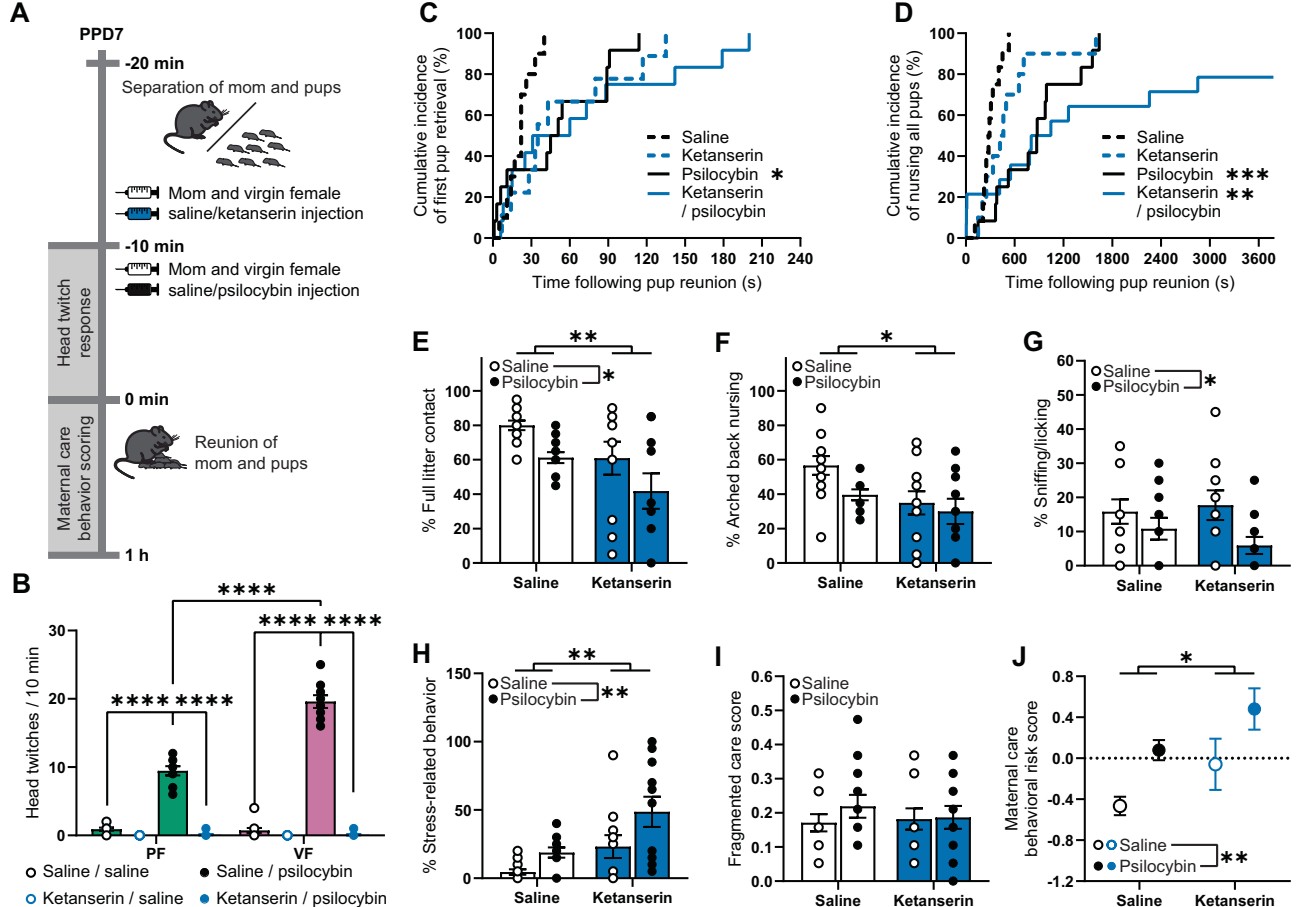

**Fig. 6 | Serotonergic agonists and antagonists acutely impair maternal care behavior. A** Timeline of experiment assessing acute maternal care behaviors in response to psilocybin and/or ketanserin as well as their pharmacodynamics at the 5-HT2A receptor. **B** Head-twitch response data comparing the effects of psilocybin and/or ketanserin in postpartum female (PF, green) and virgin female (VF, pink) mice. Data illustrated in **C–J** represent maternal behaviors exhibited by dams exposed to saline ($n = 12$), ketanserin ($n = 11$), psilocybin ($n = 12$) or ketanserin and psilocybin ($n = 11$). **C** Cumulative percentage of dams having retrieved their first pup to the nest following pup reunion. **D** Cumulative percentage of dams having initiated nursing of the entire litter for a minimum of 30 s. Percentage of observations made (**E**) in contact with the entire litter, (**F**) engaging in arched back nursing, (**G**) sniffing or licking their pups, or (**H**) exhibiting stress-related behaviors. **I** Fragmented care score indicating behavioral switching between pup-directed and non-pup-directed behaviors. **J** Integrative maternal care behavioral risk score, calculated by averaging z-normalized results from independent maternal care behaviors (arched back nursing, sniffing/licking, stress-related behaviors, and fragmented care), such that the greater the risk score, the greater the global impairment. Individual data in **B** are presented as aligned dot plots and as bar charts representing mean ± SEM with statistical comparisons between groups made using a three-way analysis of variance followed by Šídák's multiple comparison test. Data in **C–D** are presented as a survival plot and comparisons between each treatment group and saline are made using a Log-rank Mantel-Cox test with a Bonferroni correction for multiple comparisons (three total tests). Individual data in **E–I** are presented as aligned dot plots and bar charts representing mean ± SEM. Given that there are a discrete number of observations in the 1 h period, the results from multiple dams may be represented by a single point. Data in **J** are presented as mean ± SEM. Statistical comparisons between experimental groups for data in **E–J** are made using a two-way analysis of variance. *$p < 0.05$, **$p < 0.01$, ***$p < 0.001$, ****$p < 0.0001$. Exact $P$ values are provided in the Source Data.

antagonist pre-treatment exacerbated disruptive effects[34]. Thus, the maternal brain appears to be particularly sensitive to 5-HT2 receptor modulation. The dramatic effect size of ketanserin-induced maternal care deficits, particularly when co-administered with psilocybin, could be explained by a shift in the binding profile of 5-HT receptor agonists to other, non-5-HT2 receptor subtypes, as has been described in the context of cognitive flexibility[37].

Alternatively, 5-HT2Rs may play a role in the targeted restructuring of the brain during the postpartum period. For example, 5-HT2A and 5-HT2C expression are selectively upregulated in hypothalamic regions that regulate maternal care but downregulated in mesocorticolimbic circuits in postpartum rats[32]. Therefore, psilocin-induced activation of 5-HT receptors in regions of the brain that should be hypo-responsive in the postpartum period may disrupt postpartum behavior. In support of this idea, cortical microinjection of a 5-HT2A agonist blocks maternal care in rats whereas 5-HT2A agonist injections

into the medial preoptic area of the hypothalamus have no adverse effects[34]. Alternatively, the downregulation of serotonin receptors in the cortex of postpartum mice may alter the circuit activity of psilocybin, directing it toward activity in other brain regions. Indeed, psilocybin exposure increases the activity of brain regions critically implicated in parenting behavior such as the medial preoptic area, the lateral septum, and the lateral habenula[38–40] as measured by increases in c-Fos expression[41]. Interestingly, the lateral habenula has recently been identified as a region that is sensitive to maternal stress[42] and specifically activated during pup distress calls[43]. If hyperactivation of these regions contributes to maternal disruption, it is possible that a lower dose could mitigate the adverse effects of psilocybin, although a thorough dose-response study is essential to understand if this could also address its lack of efficacy. Furthermore, comparing the effects of psilocybin to that of zuranolone, the only currently marketed medication indicated specifically for the treatment of postpartum

depression, would provide useful mechanistic insight into the lack of efficacy of psilocybin in this model.

Another explanation for the differential effects of psilocybin in virgin and postpartum female mice is that the postpartum dams were exposed to daily, predictable stress in the form of pup separation and/or the psychosocial stress model, while the virgin females were not. Recent studies have demonstrated that while acute, transient corticosterone elevations due to stress (or direct exposure) are necessary for any anxiolytic effects of psilocybin, chronic (28-day) corticosterone exposure prevented psilocybin-induced anxiolysis[44]. While the present study was carried out on a far shorter timeframe, psychosocial stress and pup separation are known to increase corticosterone[45]. Future studies comparing virgin and postpartum females in comparable stress paradigms will help determine whether the differences in psilocybin responsivity observed herein are due to physiological differences across reproductive status, or differences in stress-response/-resolution, or both.

Maternal–infant interactions during the postpartum period directly guide the neurodevelopment of pups and can cause pronounced effects on emotional, social and cognitive functions lasting well into adulthood[46–48]. We showed that postpartum exposure to psilocybin induced both acute and lasting disruptions in maternal caregiving behavior. These behavioral consequences of dam psilocybin exposure had clear negative consequences on offspring. Not only did offspring of psilocybin-exposed dams show anhedonic phenotypes in the sucrose preference test, but they also had dramatically increased behavioral risk across the emotional-social-cognitive test battery regardless of sex. Pups exposed directly to psilocin by i.p injection also exhibited an anhedonic phenotype, however overall increases in behavioral risk did not reach statistical significance. From studies we have previously conducted in zebrafish, larval exposure to psilocybin/psilocin only had minimal neurotoxic effects compared to other drugs of abuse[49]. However, it is possible that psilocin exposure during breastfeeding combined with psilocybin-induced disrupted maternal caregiving behavior are necessary to adversely affect pup neurodevelopment.

There were also more nuanced sex differences observed in the adult offspring behavioral battery. In general, female offspring were at greater behavioral risk than male offspring regardless of the experimental group designation of their mothers. While there were no statistically significant sex nor interaction effects, it appears that this effect was mostly driven by the observation that female offspring reared by stressed dams were already at greater risk for behavioral deficits, and the cumulative effect of maternal psilocybin- and psychosocial stress-exposure had the most pronounced impact. The trend for the males looks distinctly different such that offspring only displayed global behavioral deficits when their mothers were exposed to both stress and psilocybin during the postpartum period. These data were in part corroborated by the larger effect size of direct psilocin exposure-induced behavioral deficits in females compared to males, and align with the clinical finding that women are twice as likely to suffer from mood disorders, which emerge at an earlier age, are more severe, and last longer compared to men[50].

Despite the therapeutic promise of psychedelics and related compounds, only a handful of studies have investigated their impact in the postpartum period. In the clinic, ketamine has been administered following cesarean section with the rationale that ketamine might serve as a prophylactic for postpartum depression[51]. Although the number of investigations is small (15), several meta-analyses examining data across 10–14 studies revealed that ketamine (racemic or esketamine) was consistently effective in the early postpartum period[52–54]. Subgroup analyses suggest that effective doses are at least 0.5 mg/kg, esketamine is more effective than ketamine, and treatment efficacy may depend on peripartum stress, with the best effects among moderately (but not mildly or severely) stressed patients[51].

Preclinically, ketamine treatment following pup weaning reduced depressive-like behaviors in a chronic stress model[55], while chronic ketamine treatment during the postpartum period produced antidepressant effects at the detriment of maternal care behavior[56]. Meanwhile only one study has assessed the consequences of postpartum serotonergic psychedelic exposure. A single injection of the psychedelic phenethylamine 25H-NBOMe, administered to rat dams on PPD 5, produced an acute dose-dependent disruption in several aspects of maternal care including pup retrieval, nursing, and pup grooming[57]. The authors found no effects on anxiety-like behavior in the open field test, although this test was conducted in the early postpartum period (PPD 5), rather than two weeks later as in the present study. Juvenile male (but not female) offspring reared by 25H-NBOMe treated dams also preferred to spend time with a familiar rather than novel mouse in the social interaction test much like we saw here. These data suggest that the adverse effects of developmental exposure to serotonergic psychedelic drugs might be conserved across rodent species. Although it is unclear whether male offspring were directly exposed to 25H-NBOMe via breast milk, there were no alterations in maternal care past the acute impairment on PPD 5, which suggests that lasting effects in offspring were likely related to direct exposure.

While our rodent model of postpartum depression possesses face and construct validity and may be useful for modeling various aspects of the disease, it is possible that it is not well suited to assess the efficacy of psychedelic compounds given that these molecules induce uniquely human subjective experiences. Moreover, the long-lasting detrimental behavioral effects measured in the offspring of dams exposed to psilocybin can likely be mitigated in humans by cessation of breastfeeding for an appropriate length of time. However, the lasting detrimental effects of psilocybin exposure during the postpartum period that we observe is in direct contrast to the lasting therapeutic effects of psilocybin exposure to nonparous females, effects that have been observed in preclinical and clinical studies.

Altogether, the data presented here suggest that while psilocybin has been consistently shown to be safe and effective for treating depression in the general human patient population, as well as in preclinical models of mental health disorders, the same may not be true for exposure during the postpartum period. The postpartum period remains an incredibly understudied aspect of human physiology, making postpartum people amongst the most vulnerable of demographics to unexpected adverse events in response to prescription or experimental new drugs. Peripartum mood disorders such as postpartum depression are in urgent need for novel, safe, effective and fast-acting therapeutics, and our study underscores the need to closely monitor behavioral outcomes in birthing parents and their offspring to establish the safety and efficacy of administering psychedelics during the postpartum period.

## Methods

### Animals and housing

C57BL/6 J (B6) mice were used for all experiments. Mice were maintained on a 12/12 light/dark cycle (lights on at 0300 h) and given access ad libitum to food and water. All behavioral testing was conducted during the active (dark) phase, 1 h after lights off, under dim red light. All behavioral coders were blind to drug treatment conditions throughout all experiments. For maternal behavior coding, experimenters administered the stress or control paradigm within minutes of data collection in distinct vivarium rooms and therefore were not blind to stress/control treatment. All procedures were conducted in compliance with the University of California, Davis Institutional Animal Care and Use Committee. Adult virgin female and male mice were obtained from Jackson Laboratories (Stock # 000664) at six weeks of age. Male intruders were virgin B6 mice, at least 60 days of age, bred in our colony and maintained in a separate vivarium room from

experimental females. Four independent cohorts of nulliparous female mice ($N = 49$) were harem bred with stud males for 17 days, after which subjects were randomly assigned to one of two model groups: control ($N = 22$) or social stress ($N = 27$) and transferred to two separate vivarium rooms. All females were then transferred to a modified home cage containing Beta Chip bedding (Newco, CA), a nestlet, food and water ad libitum, and monitored daily for the presence of pups (designated postpartum/postnatal day 0). On postpartum day 3, an additional modified cage was attached to the home cage, referred to as the escape cage. The escape cage was designated as the cage without the dam's nest and litter but was otherwise identical to the home cage. The modified caging was identical to standard shoebox caging except that modified cages included a PVC portal (4 cm diameter) that connected the two shoebox cages. Access to each compartment was controlled using a PVC pipe cap. Dams and litters were weighed daily from PPDs 4–8 at the start of behavioral testing (Supplementary Table 1).

To examine the effects of psilocybin on maternal care behaviors, control and social stress groups were further assigned to psilocybin or saline treatment groups on PPD 7 by an experimenter who was never present during maternal behavior testing. As pregnancies were not timed and mice gave birth on different dates, group assignment was pseudo-randomized to balance the treatment-assignments on a given day. The final $2 \times 2$ study design consisted of the following groups: control + saline ($N = 11$), stress + saline ($N = 16$), control + psilocybin ($N = 11$), and stress + psilocybin ($N = 11$). On PPD 9, dams and their litters were transferred to a clean, standard shoebox cage. Offspring were weaned at PPD 21 and housed in same-sex groups until adulthood (87–150 days old) before behavioral testing. Offspring were tested in 7 independent cohorts by 8 experimenters. No more than one male and one female offspring from each litter were used for behavioral testing. After weaning, all dams were exposed to a behavioral test battery to assess cognitive and emotional behavior from PPDs 23–28.

To determine whether offspring were directly exposed to psilocybin or psilocin via breast milk after maternal psilocybin treatment, a separate group ($N = 6$) of nulliparous mice were harem bred as described above and allowed to deliver pups in standard shoebox cages. On PPD 6, litters were separated from dams and kept warm on a heating pad set to low for 2 h. All dams were injected with saline immediately before pup reunion and acute maternal behavior was recorded for 1 h. On PPD 7, the procedure was repeated, however all dams were injected with psilocybin (2 mg/kg i.p.). Brain tissue was collected from at least 5 independent litters at 0 min (just before reunion), 30 min, 60 min, or 120 min after the initiation of nursing at reunion (~5 min after reunion). Pups were euthanized by decapitation and brain tissue from two pups was pooled for each sample. The dams ($N = 5$), as well as virgin female mice ($N = 6$) treated with psilocybin 2 h prior, were euthanized by cervical dislocation, and their brains removed, flash-frozen, and stored at −80 °C until analysis.

To determine whether the acute effects of psilocybin on maternal care behaviors were dependent on engagement with the serotonin system, a separate group ($N = 45$) of nulliparous mice were harem bred as described above and allowed to deliver pups in standard shoebox cages. Dams were further assigned to one of the following treatment groups to enable a $2 \times 2$ study design on PPD 7 by an experimenter who was never present during maternal behavior testing: saline + saline ($N = 12$), ketanserin + saline ($N = 11$), saline + psilocybin ($N = 12$), and ketanserin + psilocybin ($N = 11$). Adequate blocking of the acute effects of psilocybin by ketanserin were validated by a head-twitch response assay and compared to yoked virgin females.

To assess the relative expression of genes encoding serotonin receptors that have a high affinity for psilocin, a separate group ($N = 6$) of nulliparous mice were harem bred as described above and allowed to deliver pups in standard shoebox cages. Postpartum mice alongside age-matched virgin male ($N = 6$) and virgin female mice ($N = 6$) were

euthanized by cervical dislocation and brains rapidly excised. Cortical tissue was carefully removed from the rest of the brain tissue, placed into RNAse free microcentrifuge tubes, flash-frozen in liquid nitrogen and stored at −80 °C until analysis.

To assess the effects of direct psilocin exposure on PPD 7 on adult behaviors, a separate group ($N = 14$) of nulliparous mice were harem bred as described above and allowed to deliver pups in standard shoebox cages. On PPD 7, litters were separated from dams and kept warm on a heating pad set to low. No more than two males and two females from each litter were used for experiments, enabling a $2 \times 2$ study design of the following groups: male + saline ($N = 14$), male + psilocin ($N = 14$), female + saline ($N = 12$), and female + psilocin ($N = 12$). Mice were injected with psilocin slowly for approximately 1 min, then the needle held in the intraperitoneal space for another 1 min to ensure adequate absorption, before being returned to the heating pad. Mice were observed for an additional 15 min before reunion.

### Drugs
Psilocybin and psilocin were synthesized in-house and dissolved in saline to yield a final dose of 2 mg/kg[58]. As we were restricted to assessing just a single dose of psilocybin in this paradigm due to feasibility, dose selection was based on literature precedent[19,59,60]. Ketanserin tartrate (Sigma-Aldrich) was dissolved in saline to yield a final salt-corrected dose of 1 mg/kg ketanserin. Dose was selected based on experience and pilot data ensuring minimal motoric impairments. All drugs were administered to adults at a 5 mL/kg injection volume using a 28-gauge needle. Psilocin or saline was administered to pups at a 5 mL/kg injection volume using a 5/16", 31-gauge needle. All drugs were administered by intraperitoneal injection (i.p.).

### Maternal test paradigm
**Postpartum social stress model and maternal behavior assessment.** The resident intruder social stress paradigm along with subsequent maternal behavioral testing was carried out on PPD 5-7 and psilocybin or saline was administered 2 h after the third intruder exposure on PPD 7. All mice were given a brief maternal behavior pre-test on PPD 4 for habituation. On each subsequent test day, all subjects were separated from their pups for 15 min. The test procedure was identical for control and stress groups with 2 exceptions: stress mice had 75% of their nest material removed prior to the start of social stress exposure and they were repeatedly exposed to a social threat: a virgin male intruder from PPD 5–7 for 7 min during the 15 min separation. At the start of the mother–infant separation, the PVC cap was used to enclose all dams in the cage containing their nest. Dams were left undisturbed for 5 minutes, followed by a 7 min intruder or mock intruder exposure (brief exposure to the experimenter's hand to simulate the placement of another mouse). During the last 3 min of the separation, the PVC cap was removed so that dams could access both cages freely.

At reunion, pups were scattered in the nest cage. Latency data were collected exclusively for 3 min and maternal behavior was recorded for the following hour by trained observers. Latency data were recorded for the following: retrieval of first pup, retrieval of all pups to nest, and adoption of a nursing posture over all pups for at least 30 s. The frequency of pup retrievals that involved circling behavior were also recorded. Circling behavior occurred when dams picked up displaced pups and moved them around the cage to locations other than the nest or future nest (in the case of nest relocation). During circling, dams typically pick up pups and put them down in rapid succession, sometimes running from back and forth between cages with a pup in their mouths. The frequency of nursing, nest-building, nest-related and non-litter related behaviors (a total of 24 behaviors) were then scored every 3 min for 1 h and the cage locations of the dam and pups were noted.

One hour after the end of behavioral testing on intruder day 3 (PPD 7), all dams were injected with either saline or psilocybin (2 mg/kg i.p.) by a separate experimenter. On PPD 8 all dams received a post-test identical to the behavioral paradigm administered on PPDs 5–7, except that no one was exposed to a male intruder during the 15 min separation. On PPDs 11–12, all dams were tested for sucrose preference.

**Post-wean maternal assessment.** Once offspring were weaned, dams remained singly housed and were exposed to a small battery of standardized tests to assess cognitive, anxiety-related, self-care and depressive-like behaviors beginning on PPD 23, specifically selected as classical ongoing features of PMDs. Cognitive behavior was assessed using the novel object recognition test (NOR), anxiety-related behaviors were assessed using the open field test and elevated plus maze, self-care behavior was assessed using a sucrose splash test, and emotional behavior was assessed using the forced swim test, all of which are described below. The battery order was determined by increasing degrees of stress to the animal, in compliance with the University of California, Davis Institutional Animal Care and Use Committee guidelines. This test battery began 16 days after psilocybin treatment on PPD 7. Virgin female mice were also subjected to the same battery 16 days following a single dose of psilocybin to see how psilocybin exposure during the postpartum period may compare to non-postpartum exposure.

**Adult offspring assessment**

Once offspring were weaned and aged to adulthood (minimum 12 weeks old), they were exposed to a larger battery of standardized tests to assess multiple functional domains that could be relevant to several neuropsychiatric disorders; social, emotional and cognitive behavior (open field test, novel object recognition, T-maze spontaneous alternation, social interaction, sucrose preference, forced swim test). This test battery began at least 11 weeks following maternal psilocybin exposure on PPD 7. The battery order was determined by increasing degrees of stress to the animal, in compliance with the University of California, Davis Institutional Animal Care and Use Committee guidelines.

**Sucrose preference.** For dams, habituation to two water bottles began on PPD 10, followed by a 24 h sucrose preference test on PPDs 11–12, before the offspring were old enough to reach the sippers. For adult offspring and virgin female mice, test mice were briefly single housed for testing. All mice were habituated overnight to two sipper bottles containing water. The following day, one of the two bottles was replaced with a bottle that contained a 1% sucrose solution. The location of the replaced bottle was counterbalanced and the weight of both bottles was recorded. 24 h later, bottle weights were again recorded. Sucrose preference was calculated as: consumption$_{sucrose}$ / consumption$_{sucrose + water}$ where consumption was calculated as the final weight of the sipper bottle minus the original weight.

**Open field test.** Mice were placed in a translucent perspex open field chamber measuring 45 cm L × 45 cm W × 56 cm H and allowed to freely explore the chamber for either 10 min for dams or virgin females, or 30 min for the adult offspring. At the conclusion of the test, animals were returned to their home cages and the test chambers were cleaned with 70% ethanol between animals. Time spent in the center zone was automatically scored using ANY-maze software version 7.43 by tracking time spent in the inner 3 × 3 space of a 5 × 5 grid defining the chamber base.

**Novel object recognition.** Mice were placed in the same open field chamber as for the open field test and allowed to freely explore the chamber for 10 min to re-acclimate before being returned to their home cage and the chamber cleaned with 70% ethanol. One hour later, mice were returned to the open field chamber this time containing a pair of identical objects as a training phase; either a 50 mL conical centrifuge tube, an extra-large binder clip, a 30 mL glass conical flask, or a structure made of Lego™. After 10 min, mice were removed and placed in their home cage and the chamber and objects were cleaned with 70% ethanol. After 1 h, mice were reintroduced to the chamber for 10 min for the testing phase, but this time one object was replaced with a novel object from the object list above. The chamber location and object locations were counterbalanced across mice. ANY-maze software version 7.43 automatically tracked the amount of time interacting with the novel and familiar objects with interactions defined as time spent with their nose within 1.5 cm of the object and oriented toward the center of the object. The time spent exploring each object was used to calculate the discrimination index: $(t_{novel\ object} - t_{familiar\ object}) / (t_{novel\ object} + t_{familiar\ object})$. The chamber and objects were thoroughly cleaned with 70% ethanol between trials and mice. A minimum exploration time of 20 s for each object during the training phase was necessary for their results in the testing phase to be included in analysis. As an insufficient number of adult offspring reached criteria in the training phase, the NOR results were excluded from total analysis.

**Elevated plus maze.** The elevated plus maze apparatus consisted of a white plus-shaped perspex platform positioned 50 cm off the ground. All four arms measured 40 cm L × 7 cm W, with two opposite arms being enclosed by 25 cm high walls and the other two arms being open with unprotected edges. Mice were placed at the center of the maze facing a closed arm and allowed to explore freely for 5 min. Mice were returned to their home cages and the apparatus cleaned with 70% ethanol. Noldus Observer XT 12.5 was used to manually code the amount of time spent in the closed and open arms. An animal entered a closed or open arm when all four paws were in the arm.

**Sucrose splash test.** A 10% sucrose solution was prepared in a spray bottle. At the start of the test, 4 sprays were delivered to the animal's dorsal coat in the home cage and behavior was video recorded for 5 minutes. Grooming behavior (licking, scratching and/or face washing), which is considered an indication of self-care[61], was manually quantified by trained observers blinded to the treatment group using Noldus Observer XT 12.5.

**Forced swim test.** Mice were placed in translucent perspex cylinders measuring 18 cm D × 22 cm H, filled with 15 cm of 24 +/− 1 °C water, for 6 min before being dried and returned to their home cage. Following a 2 min habituation period, the final 4 min of each video was scored for immobility by trained observers blinded to the group assignment of the animal, manually quantified using Noldus Observer XT 12.5. Immobility was considered no movement (floating) or minor movements necessary to maintain afloat.

**T-maze spontaneous alternation task.** Mice were placed in the long arm of a T maze [61 × 11.4 × 8.3 cm] facing the central node and allowed to freely explore for 10 min. The other two other arms, B & C [33.7 × 11.4 × 8.3 cm], were separated such that entry between them necessitated entry into the long arm of the apparatus. Animal movement was analyzed using ANY-maze software version 7.43 and percent alternation calculated by: #alternations$_{B\&C}$ / (#entries$_{B + C}$ − 1) × 100.

**3-chamber social interaction test.** The social interaction apparatus was made of translucent perspex and consisted of three identical chambers measuring 40 cm L × 20 cm W × 22 cm H each separated by a central door that could be removed to allow free access between the chambers. The two outer chambers housed a cage for holding another mouse, or an object. Subject mice were placed in the center chamber and gates removed to allow free access to the apparatus and empty

cages for 10 min. Mice were then removed, gates replaced, and a novel object was placed in one cage and a littermate mouse placed in the other. Subject mice were then replaced in the center chamber, gates removed, and mice allowed to interact with the social animal or the object to assess sociability. Mice were then removed, gates replaced, and the object replaced with a novel mouse. The position of the novel mouse and littermate were swapped. Subject mice were then replaced in the center chamber, gates removed, and mice allowed to interact with their littermate or the novel mouse to assess preference for social novelty. The time spent in each of the interaction chambers was manually quantified using Noldus Observer XT 12.5 and used to calculate either the sociability index: $(t_{mouse} - t_{object}) / (t_{mouse} + t_{object})$, or the social novelty preference index: $(t_{novel\ mouse} - t_{familiar\ mouse}) / (t_{novel\ mouse} + t_{familiar\ mouse})$. Chambers were thoroughly cleaned with 70% ethanol between subjects.

**Head-twitch response.** A psilocybin dose-escalation study was carried out on age-matched virgin male, virgin female, and parous female mice. Across three weeks, mice received weekly escalating doses starting with a saline injection, then 0.5 mg/kg, 2 mg/kg, and 8 mg/kg psilocybin, before being placed in an empty cage with no lid and filmed from the top for 10 min. For the ketanserin + psilocybin experiment, yoked postpartum and virgin female mice were administered a dose of saline or 1 mg/kg ketanserin and singly-housed in a temporary holding cage for 10 min. Mice were then dosed with saline or 2 mg/kg psilocybin and immediately placed in an empty cage with no lid and filmed from the top for 10 min. Cages were cleaned with 70% ethanol between experiments. At the conclusion of the experiments, each video was manually scored for the number of head-twitches using Noldus Observer XT 12.5 by trained observers blinded to the treatment.

## Metabolomics

**Sample extraction.** Mice were euthanized by cervical dislocation and rapid decapitation. Brain tissue was rapidly extracted, flash frozen with liquid nitrogen and stored at −80 °C for later use. 100 mg brain was homogenized using a GenoGrinder 2010 (SPEX SamplePrep) in 1.5 ml ice-cold acetonitrile/methanol/water (2:2:1, v/v/v) with 37.5 µl 0.1 M ascorbic acid and 3 ng bufotenine-D4, at −20 °C with three 3 mm metal beads. The tissue was homogenized three times for 20 s at 1500 rpm, with a 30-second interval between homogenizations. The homogenate was then extracted and sonicated in an ice bath for 15 min, followed by centrifugation for 2 minutes at $14,000 \times g$ to precipitate proteins. The resulting supernatant was transferred and evaporated to dryness using a CentriVap centrifugal vacuum concentrator (Labconco). The dried extracts were reconstituted in 140 µl of methanol:water (1:1, v/v). After centrifugation for 2 minutes at $14,000 \times g$, the sample was transferred to a high-performance liquid chromatography (HPLC) vial with an insert and was ready for analysis.

**Instrumental method and Quality Assurance/Quality Control (QA/QC).** Instrumental analysis was performed using a Vanquish UHPLC system coupled to a SCIEX QTRAP 6500 + LC-MS/MS system (SCIEX). A Thermo Scientific Accucore Biphenyl Reverse Phase HPLC column (particle size 2.6 µm, i.d. 2.1 × length 100 mm) coupled with Accucore Defender Guard Column (particle size 2.6 µm, i.d. 2.1 ×10 mm). The mobile phase comprised of 5 mM of ammonium acetate in HPLC water (A) and 100% Methanol (B). A gradient at a flow rate of 0.3 ml/min and column temperature was 40 °C. The linear gradient was set as follows: 0 to 1.5 min: 4% B; 1.5 to 2.5 min: 4% B to 20 % B; 2.5 to 5.5 min: 20% B to 40% B; 5.5 to 6.5 min: 40% B to 90% B; 6.5 to 7.5 min: 90% B (hold for 1 min); 7.5 to 8.5 min: 90% B to 4% B, with 2.5 min used as a post-run equilibrium. Other parameters were set as follows: temperature 500 °C, curtain gas 35 psi, ion spray voltage 2500 V, nebulizer gas: 50 psi and heater gas: 40 psi. The MS/MS was operated in Scheduled MRM Pro Algorithm in positive polarity mode. Parameters were

optimized by infusion of individual analyte as shown in Supplementary Table 2. Phosphate-buffered saline was used as procedural blanks. A QC sample was prepared by pooling all treated and control samples. QC analysis was conducted once for every six injections of biological samples and a blank sample (methanol:water, 1:1, v/v). A nine-point calibration curve was prepared by spiking 3 ng bufotenine-D4 and varying amounts of psilocin and psilocybin (0.02, 1.5, 2.5, 5, 10, 20, 30, 40 and 50 ng in Cal1 to Cal 9). Peak picking and integration were performed using SCIEX MultiQuant 3.0 software.

**Quantitative analysis of mRNA by real-time qPCR.** Total RNA was isolated and purified using the RNeasy® Lipid Tissue Mini Kit (Qiagen 74804) and the optional on-column DNase digestion (Qiagen 79254). A Nanodrop™ Spectrophotometer was used to determine the quality and quantity of the RNA. All samples were high quality (260/280 - 2.0). The cDNA templates were prepared from 1 µg of RNA using the iScript™ cDNA Synthesis Kit (Biorad 1708890). Quantitative real-time PCR was performed using the QuantStudio™ 7 Pro (Applied Biosystems). PCR products were detected using the following TaqMan® Gene Expression assays: *Htr1a* (Mm00434106_s1), *Htr2a* (Mm00555764_m1), *Htr2c* (Mm00434127_m1), and *Bdnf* (Mm04230607_s1). All samples were normalized to beta-2 microglobulin (*B2m*, Mm00437762_m1). Target and endogenous control genes were measured in triplicate on a single 384 well plate. One sample was omitted due to high variability across replicates (standard deviation > 0.3). There were no statistically significant differences in the expression of the endogenous control gene between groups. Relative quantification was determined by QuatStudio™ Design and Analysis Software (2.8.0) using the comparative cycle thresholds (delta delta Ct) method.

## Statistical analysis

All data were analyzed using GraphPad Prism (v10.5). Maternal behavior composite scores were calculated by summing the total number of observations in which the subject was engaged in a behavior or set of behaviors in physical contact with pups and dividing by the total number of possible observations (20) multiplied by 100. Fragmented care was calculated by summing the number of observations where the dam transitioned between behavior groups (nursing, nest-building, nest-related and non-litter related) out of 19 possible transitions[15,62]. The maternal stress-related behavior score represents a sum of behaviors that diverge from the stereotyped pattern of caregiving described in rodents. These behaviors include partial litter contact (contact with some but not all pups), abnormal in-nest behaviors (out of pup contact or contact that prevents pups from accessing nipples), or stress-related behaviors outside of the nest such as digging, and self-grooming or inactivity (Supplementary Data 1). Latency data were analyzed using a Kaplan-Meier method whereby median latencies to retrieve the first pup across the three intruder days were plotted using a cumulative incidence plot (simple survival analysis in GraphPad Prism), and a Log-rank Mantel–Cox test used to make statistical comparisons between groups[63] followed by a Bonferroni multiple comparisons test when significant. Pup circling and nest relocation data were analyzed using contingency tables, and statistical comparisons between groups made using a Chi-square (and Fisher's exact) test.

All other maternal behaviors and non-maternal dam behaviors were analyzed using 2way ANOVAs either comparing the effects of stress across the three intruder days, comparing the interaction between stress and psilocybin exposure or comparing the interaction between ketanserin and psilocybin exposure on behavioral outcomes. All manual behavioral coding was conducted by independent raters trained to >90% agreement during practice sessions that were blind to group assignment.

Virgin female behavioral battery test data were analyzed using t-tests comparing saline- to psilocybin-treated animals. Yoked virgin female and postpartum female head-twitch response data were

analyzed using a 3way ANOVA comparing the effects of reproductive state, ketanserin, and psilocybin exposure, followed by Šídák's multiple comparisons test comparing the means of groups that differed by only one factor.

All offspring behavioral data were analyzed either using 3way ANOVAs comparing the effects of maternal stress, maternal psilocybin exposure, and offspring sex on behavioral outcomes, or 2way ANOVAs comparing the effects of psilocybin exposure and offspring sex on behavioral outcomes. Significant interaction effects involving sex from 3way ANOVAs were followed up with 2way ANOVAs split by sex to determine the nature of the sex differences. For the psilocybin dose-escalation headtwitch response experiment, groups were analyzed by 2way ANOVA, followed by planned comparisons between postpartum females and all other groups. Given that no mice exhibited headtwitches in response to saline, these data were removed from analysis to prevent a forced interaction effect between dose and group. The relationship between pup avoidance and sucrose preference was analyzed using a two-tailed Pearson correlation analysis.

Behavioral battery data for postpartum females, virgin females, or offspring were also analyzed using an integrative technique to calculate a "behavioral risk score", whereby a higher score indicates more behavioral impairments across the complete battery of tests. Results from each test were first converted to z-scores following removal of any statistical outliers. Z-scores are standardized scores indicating how many standard deviations the result is above or below the mean of all observations made in the test, using the following equation: $z = (X − μ)/σ$, where $z$ is the standardized score, $X$ is the measured result in the behavioral task, and μ and σ are respectively the mean and standard deviation of all results measured in the behavioral task. To calculate the integrated behavioral risk score, directionality was established for each behavior such that higher z-scores indicated behavioral impairments specific to each task. For example, z-normalized data for results in the T-maze were multiplied by −1 such that mice that performed fewer alternations, and therefore showed worse working memory, were attributed a higher z-score indicative of this impairment. Immobility in the forced swim test, however, is associated with an increased depressive-like phenotype and therefore z-normalized immobility measures were multiplied by 1. Finally, sociability and social novelty scores were first averaged before integration with other tests to ensure equal weighting across behavioral domains. An overall behavioral risk score was then calculated for each mouse by taking the average of z-normalized results across all tests: Behavioral risk score = $\frac{Z_{Test1} + Z_{Test2} + Z_{Test3} + \cdots}{Number\ of\ tests}$, where $Z_{Test\,1}$ etc. is the average z-score of all animals in that test[64].

Similarly, a maternal care behavioral risk score was calculated by averaging z-normalized results for independent maternal care behaviors; arched back nursing, sniffing/licking, avoidance (when available), stress-related behaviors and fragmented care. Note that the maternal behavior risk score calculated for the experiment examining psilocybin/psilocin transfer to offspring does not include an avoidance measure as these mice were housed in standard cages.

Effect sizes were calculated for all behavioral risk scores; η² values were calculated for ANOVAs where values around 0.01 were considered small, 0.06 considered medium, and 0.14 considered large[65], and Cohen's d values were calculated for t tests where values around 0.2 were considered small, 0.5 considered medium, and 0.8 considered large.

Relative quantification values were calculated and normalized by log₂-transformation for each animal and each transcript according to the following formula:

$$\log_2 \text{fold change} = \log_2(2^{−ΔΔCt})$$

where ΔΔCt is the Ct of a given transcript, subtracted from the Ct of the housekeeping gene (beta-2-microglobulin) of the same animal, and then subtracting from that the average Ct values of the given transcript from the reference animals (virgin females).

In all instances, statistical outliers were determined by a ROUT test with Q = 1%. Normal distribution and sphericity were assumed for all ANOVA data. Data were considered statistically significant if $p < 0.05$.

## Reporting summary
Further information on research design is available in the Nature Portfolio Reporting Summary linked to this article.

## Data availability
Source data are provided with this paper. In addition, the raw data for this manuscript are available on Figshare (https://doi.org/10.6084/m9.figshare.26376361). Source data are provided with this paper.

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

## Acknowledgements

We would like to thank Dr. Hunter Warren and Anna Vernier for the synthesis of drugs used in this study. We would like to thank Max Kymer, Yash Prasad, Paris Kindle, Rachel Fruchtmann, Rayna Fukumoto, Avani Nagpal, and Jacob Sommer for assistance in generating behavioral data. This work was supported by funding from the National Institutes of Health (R01HD087709, D.S.S.; R35GM148182, D.E.O.), the W. M. Keck Foundation (D.E.O.), the University of California at Davis Pilot Project Program Award from the Perinatal Origins of Disparities Center (D.S.S.), and the University of California at Davis Academic Senate Large Grant Award (D.S.S.).

## Author contributions

C.J.H., D.E.O. and D.S.S. designed research; C.J.H., M.L., A.L., S.J.L., H.M., S.V., E.A., L.B., M.M.R., N.J., M.K., C.C., Y.A.K. and D.S.S. acquired data; C.J.H., M.L., O.F., D.E.O and D.S.S. analyzed and interpreted data; C.J.H, D.E.O. and D.S.S. wrote the paper.

## Competing interests

D.E.O. is a cofounder of Delix Therapeutics, Inc., serves as the Chief Innovation Officer and Head of the Scientific Advisory Board, and has sponsored research agreements with Delix Therapeutics and Reunion Neuroscience. Delix Therapeutics and Reunion Neuroscience were not involved in the conceptualization, design, decision to publish, or preparation of this manuscript. The remaining authors declare no competing interests.
