## [Transparent Peer Review file · Nature Communications]

Psilocybin during the postpartum period induces long-lasting adverse effects in both mothers and offspring

Corresponding Author: Professor Danielle Stolzenberg

Version 0:

Reviewer comments:

Reviewer #1

(Remarks to the Author)

This manuscript describing the behavioral outcomes in pregnant mice and their offspring following exposure to psilocybin is a well-written and timely manuscript that demonstrates notable changes in anxiety and threat-sensitivity related behaviors following material exposure to the compound. Notable strengths of the study include the use of multiple behavioral tests to assess behavior, inclusion of virgin control animals, and quantification of brain concentrations of psilocin in parent and offspring animals. Notable areas of concern include the lack of a dose-response element in the study design, repeated exposures of single animals to multiple stressful behavioral paradigms in a single battery without order randomization, use of non-matched batteries to assess mood-disorder related behavioral phenotypes between parous mice and their adult offspring, and lack of mechanistic investigation regarding the source of these behavioral changes.

Overall, these observations are of important relevance to the field, but their impact would be improved with the inclusion of additional experimentation and discussion to address the following specific points of consideration:

- 1) While 2 mg/kg psilocybin is a reasonable dosage choice given the previous preclinical investigations in mice, it is not clear whether the effects being reported are generalizable across other dosages. An experiment such as the one reported in figure 2 that demonstrated dose-responsiveness would increase the confidence in the attributed pharmacological basis for these findings.
- 2) Minor - the title for figure 2 implies that psilocybin is being used as a protective intervention against social stress, but this wording seems awkward, given that the drug is being provided only after the stress has been applied.
- 3) The effect of the stress paradigm on sniffing and licking behaviors appears inconsistent between tested cohorts (see Fig 1G and 2E), and commentary on the robustness of this and other sub-measures would be helpful for the reader.
- 4) The differential effects of psilocybin on postpartum versus virgin mice are quite interesting, and may point to an important impact of prior and ongoing stress, as well as stress-resolution, to the behavioral consequences of psilocybin exposure. The ongoing stressful nature of the behavioral battery is also relevant to this issue, and recognition of possible order-effects would be appropriate. Further, the topic has recently been explored in the literature and could be a useful area for further discussion. See, for example, doi: 10.1021/acspsci.3c00123
- 5) Commentary on why different behavioral batteries were chosen for parous versus offspring animals would be helpful.
- 6) Additional experiments to understand whether the psilocybin-induced effects are dependent on serotonergic activity (for example, see doi: 10.1016/j.isci.2024.109686.), HPA axis modulation, both, or neither would be quite helpful and add important mechanistic value to the behavioral observations already reported in this manuscript.

Reviewer #2

(Remarks to the Author)

This is an excellent study by Hatzipantelis et al. to test the potential use for psilocybin in a mouse model of postpartum depression. Contrary to prediction that psilocybin may be beneficial for PPD, their results show quite convincingly that

psilocybin may have negative consequences in this mouse model. This is an important result that should be reported and will provide a sharp contrast to the mostly beneficial results seen in control adult animals and humans. The study is impactful and has clinical relevance because there are ongoing clinical trials testing psilocybin for postpartum depression in humans.

The strength of the study is that the behavioral characterizations are carried out with a decent size cohort and thus well powered. The figures were prepared meticulously, and it is easy to see the timeline of the full experiment. The writing was clear. I have only a few comments that may be helpful.

Comments

Does this mouse model of PPD have predictive validity? Do other known treatment for PPD such as brexanolone have the expected beneficial effect if administered and tested using this mouse model of PPD?

It may be helpful for future studies to note that brain regions implicated in parenting behavior (PMID: 27921311), such as medial preoptic nucleus, auditory cortex, and lateral septum, exhibit psilocybin-evoked change in cFos expression (PMID: 36630309), which may be potential areas mediating the behavioral differences in maternal care.

Are the later behavioral deficits in pups in this study seen in Figure 4 due to the reduced maternal care and/or the psilocin exposure in the pup? Can these contributions be isolated and tested?

Reviewer #3

(Remarks to the Author)

The manuscript by Hatzipantelis et al describes a series of experiments in mice that aimed to determine whether psilocybin was effective at improving the disruptions to maternal care behaviours elicited by stress, as well as evaluating effects on anxiety-like behaviours in dams and their adult offspring. The authors employed standard batteries of behavioural test to determine these outcomes, in combination with a novel postpartum stress paradigm that had been previously validated by this group to interfere with maternal behaviour. These tests confirmed that stress exposure impaired maternal care behaviours, but that psilocybin administration did not prevent impairments associated with social stress. In fact, psilocybin administration in mouse dams during the postpartum period had long lasting adverse effects on mood related behaviours in both mothers and adult offspring.

These results represent an important new avenue of research into psychedelics as therapies, suggesting a potential risk of adverse consequences for people with peripartum mental illness. The candid report of these (possible) negative implications is particularly commendable in the current climate of extreme hype about the potential benefits of psychedelic substances across mental health diagnoses, without much attention paid to possible risks. The fact that this patient population (i.e. women with postpartum depression) are being actively recruited for clinical trials using psilocybin analogues makes this work especially timely.

The methodology, analysis and interpretation of results are sound. However, a major criticism of this manuscript is that it does not include an assessment of potential mechanisms that might be responsible for why post-partum female responses to psilocybin with increased anxiety-like behaviour, in contrast to its anxiolytic effects in virgin females. It seems plausible that the biological/molecular mechanisms that completely reverses the responses of parous females to psilocybin may relate to the reduction in head twitch responses (shown in Supplementary Fig 1). This could reflect a change in the serotonin receptor binding profile of psilocybin in parous dams (i.e. directed away from the 5-HT_{2A}) that is critical to examine in the context of a) altered sex hormone profiles compared to the virgin state and b) the potential that estrogen and serotonin receptor interactions mediate the different behavioural responses to psilocybin in parous female mice.

Furthermore, it is impossible to disentangle the effects of psilocybin (psilocin) exposure during breastfeeding on adult offspring behaviour from the effects of disrupted maternal care, which has important implications for risk evaluation. This could be examined with a cross-fostering experiment (psilocybin-treated dams foster saline-treated pups and vice versa).

I also wonder whether the authors have validated that exposure to the intruder mouse in the post-partum period elicits a physiological stress response (i.e. increased CORT)? For understanding the mechanisms that drive changes in maternal care behaviours, it is important to know whether stress hormone signalling, as opposed to pheromone release or ultrasonic vocalisations, underlies the behavioural effects of this model.

Finally, for the comparisons between postpartum and virgin females in Figure 5, all dams received saline first, then psilocybin the day after, and maternal care was evaluated within-animal as a change between saline and psilocybin. However, is it also possible that the differences observed on PPD 7 are related to the additional "stressor" of the pup separation (2 hours) and reunion on PPD 6, when all dams received saline? Counterbalancing mice to receive either saline or psilocybin first on PDD 6 and then the other treatment on PPD 7 would help to clarify this.

Version 1:

Reviewer comments:

Reviewer #1

(Remarks to the Author)

The changes the authors made, including both textual alterations and additional experiments, are responsive to the comments made by all reviewers, and the article is suitable for publication.

Reviewer #2

(Remarks to the Author)

The authors did a good job addressing the comments. I remain quite enthusiastic about the study, as it represents a considerable amount of behavioral assays, looking at potential effects of psychedelics on rodent models relevant for postpartum depression. The study is well executed with a decent number of samples per group, and is relevant when considering active clinical trials that seek to use psychedelics or related psychoactive compounds for treating postpartum depression.

Reviewer #3

(Remarks to the Author)

The authors have thoroughly and satisfactorily addressed all concerns raised in my previous review. The revisions demonstrate improved methodological rigor, clearer presentation of results, and more nuanced interpretation of findings. The additional analyses and clarifications requested have been implemented effectively, strengthening the overall quality and impact of the work. I recommend acceptance of this manuscript for publication, as it will make a valuable contribution to our understanding of the potential adverse effects of psilocybin during the postpartum period for mothers and offspring.

First, we would like to thank the reviewers for their insightful comments about our work, enabling us to make changes to the manuscript that represent a great improvement over the previous iteration. Our responses are highlighted in blue below each comment.

REVIEWER COMMENTS

Reviewer #1 (Remarks to the Author):

This manuscript describing the behavioral outcomes in pregnant mice and their offspring following exposure to psilocybin is a well-written and timely manuscript that demonstrates notable changes in anxiety and threat-sensitivity related behaviors following material exposure to the compound. Notable strengths of the study include the use of multiple behavioral tests to assess behavior, inclusion of virgin control animals, and quantification of brain concentrations of psilocin in parent and offspring animals. Notable areas of concern include the lack of a dose-response element in the study design, repeated exposures of single animals to multiple stressful behavioral paradigms in a single battery without order randomization, use of non-matched batteries to assess mood-disorder related behavioral phenotypes between parous mice and their adult offspring, and lack of mechanistic investigation regarding the source of these behavioral changes.

Overall, these observations are of important relevance to the field, but their impact would be improved with the inclusion of additional experimentation and discussion to address the following specific points of consideration:

1) While 2 mg/kg psilocybin is a reasonable dosage choice given the previous preclinical investigations in mice, it is not clear whether the effects being reported are generalizable across other dosages. An experiment such as the one reported in figure 2 that demonstrated dose-responsiveness would increase the confidence in the attributed pharmacological basis for these findings.

Change: Text

We agree that establishing a dose-response relationship for the effects of psilocybin in this psychosocial stress model would help rationalize the results seen herein. Unfortunately, given the labor-intensive nature of the paradigm, we are limited to testing four appropriately powered groups, i.e. one vehicle and one dose with and without the stress paradigm. Adding two more doses, as would be necessary for an adequate dose-response study, would require eight experimental groups running in parallel. The equipment/personnel such a study requires would preclude us from doing any other experiments at the same time and given our current resources it would likely take years to complete. As such, we selected the most optimal dose based on strong literature precedent. In the interest of transparency and acknowledgement of the value such a study would provide, we added the following caveat to the methods section: "As we were restricted to assessing just a single dose of psilocybin in this paradigm due to feasibility, dose selection was based on literature precedent¹⁷⁻¹⁹." and the following added to the discussion: "Future studies establishing the dose-response relationship of the efficacy and adverse event potential of psilocybin in both parous and non-parous female mice would help elucidate if any differences were as a result of changes in potency."

2) Minor - the title for figure 2 implies that psilocybin is being used as a protective intervention against social stress, but this wording seems awkward, given that the drug is being provided only after the stress has been applied.

Change: Text

We changed Figure 2 title to: "Psilocybin does not treat social stress-induced sustained impairments in maternal care and produces maternal care deficits itself"

3) The effect of the stress paradigm on sniffing and licking behaviors appears inconsistent between tested cohorts (see Fig 1G and 2E), and commentary on the robustness of this and other sub-measures would be helpful for the reader.

Change: Text

To be clear, Figure 1G and 2E are the same animals at different stages of the experiment (i.e., 1G is before and 2E is after psilocybin treatment, see Figures 1A and 2A). To clarify, we have modified the text of Fig. 1 and Fig. 2 legends to read:

Fig. 1.

"Timeline of experiment. Data illustrated in **B–J** represent maternal behaviors exhibited by control (n=22) and stressed (n=27) dams, prior to drug administration, measured during the 1 h observation period following intruder exposure on PPDs 5–7 (shaded in **A**)."

Fig. 2.

"Timeline of experiment. Data illustrated in **B–H** represent maternal behaviors exhibited by control dams exposed to saline (n=11) or psilocybin (n=11), and stressed dams exposed to saline (n=16) or psilocybin (n=11) measured during a 1 h

observation period 24 h following final intruder and psilocybin exposure on PPD8 (shaded in **A**). Animals used in these experiments are the same as those used in Fig. 1.”

We also added commentary throughout the results section as to the robustness compared to what has been previously observed using this model, for e.g.: “Importantly, all above observations were in accordance with previously published results using this model¹⁴.”

4) The differential effects of psilocybin on postpartum versus virgin mice are quite interesting, and may point to an important impact of prior and ongoing stress, as well as stress-resolution, to the behavioral consequences of psilocybin exposure. The ongoing stressful nature of the behavioral battery is also relevant to this issue, and recognition of possible order-effects would be appropriate. Further, the topic has recently been explored in the literature and could be a useful area for further discussion. See, for example, doi: 10.1021/acsptsci.3c00123

Change: Text

We added the following caveat to the methods section: “The battery order was determined by increasing degrees of stress to the animal, in compliance with the University of California, Davis Institutional Animal Care and Use Committee guidelines.” We also added to the discussion, commentary on the effects of ongoing stress and stress-resolution to the behavioral consequences of psilocybin exposure: “Another explanation for the differential effects of psilocybin in virgin and postpartum female mice is that the postpartum dams were exposed to daily, predictable stress in the form of pup separation and/or the psychosocial stress model, while the virgin females were not. Recent studies have demonstrated that while acute, transient corticosterone elevations due to stress (or direct exposure) are necessary for any anxiolytic effects of psilocybin, chronic (28-day) corticosterone exposure prevented psilocybin-induced anxiolysis⁴⁸. While the present study was carried out on a far shorter timeframe, psychosocial stress and pup separation are known to increase corticosterone⁴⁹. Future studies comparing virgin and postpartum females in comparable stress paradigms will help determine whether the differences in psilocybin responsivity observed herein are due to physiological differences across reproductive status, or differences in stress-response/-resolution, or both.”

5) Commentary on why different behavioral batteries were chosen for parous versus offspring animals would be helpful.

Change: Text

We chose a behavioral battery for parous animals that reflected well known features of PMDs. For their offspring, we did not know which types of behavioral deficits to expect, so we chose a more general behavioral battery relevant to a wide range of neuropsychiatric conditions. To clarify, we have added descriptions rationalizing selection for both dam and offspring behavioral batteries to the methods: “Once offspring were weaned, dams remained singly housed and were exposed to a small battery of standardized tests to assess cognitive, anxiety-related, self-care and depressive-like behaviors beginning on PPD 23, specifically selected as classical ongoing features of PMDs.” And “Once offspring were weaned and aged to adulthood (minimum 12 weeks old), they were exposed to a larger battery of standardized tests to assess multiple functional domains that could be relevant to several neuropsychiatric disorders; social, emotional and cognitive behavior (OFT, NOR, T-maze, SI, Sucrose preference, EPM, FST).”

6) Additional experiments to understand whether the psilocybin-induced effects are dependent on serotonergic activity (for example, see doi: 10.1016/j.isci.2024.109686.), HPA axis modulation, both, or neither would be quite helpful and add important mechanistic value to the behavioral observations already reported in this manuscript.

Change: Experiment

To understand whether the effects of psilocybin were due to activity at the serotonin system, we carried out maternal care behavior assessments acutely following exposure to psilocybin and/or the 5-HT₂ receptor antagonist, ketanserin. We found that ketanserin not only failed to prevent psilocybin-induced impairments in maternal care, but it itself produced maternal care impairments independently of psilocybin. We have added this study as Figure 6 of the manuscript and added relevant discussion.

Reviewer #2 (Remarks to the Author):

This is an excellent study by Hatzipantelis et al. to test the potential use for psilocybin in a mouse model of postpartum depression. Contrary to prediction that psilocybin may be beneficial for PPD, their results show quite convincingly that psilocybin may have negative consequences in this mouse model. This is an important result that should be reported and will provide a sharp contrast to the mostly beneficial results seen in control adult animals and humans. The study is impactful and has clinical relevance because there are ongoing clinical trials testing psilocybin for postpartum depression in humans.

The strength of the study is that the behavioral characterizations are carried out with a decent size cohort and thus well powered. The figures were prepared meticulously, and it is easy to see the timeline of the full experiment. The writing was clear. I have only a few comments that may be helpful.

Comments

Does this mouse model of PPD have predictive validity? Do other known treatment for PPD such as brexanolone have the expected beneficial effect if administered and tested using this mouse model of PPD?

Change: Text

We have not yet done this experiment but are actively applying for funding opportunities to assess this. To acknowledge this important limitation we have added the following caveat to the discussion section: "Furthermore, comparing the effects of psilocybin to that of zuranolone, the only currently marketed medication indicated specifically for the treatment of postpartum depression, would provide useful mechanistic insight into the lack of efficacy of psilocybin in this model."

It may be helpful for future studies to note that brain regions implicated in parenting behavior (PMID: 27921311), such as medial preoptic nucleus, auditory cortex, and lateral septum, exhibit psilocybin-evoked change in cFos expression (PMID: 36630309), which may be potential areas mediating the behavioral differences in maternal care.

Change: Text

We agree that this would be valuable and have added the following to the discussion in the context of mechanisms explaining the sensitivity of the postpartum brain to psilocybin: "Alternatively, the downregulation of serotonin receptors in the cortex of postpartum mice may alter the circuit activity of psilocybin, directing it toward activity in other brain regions. Indeed, psilocybin exposure increases the activity of brain regions critically implicated in parenting behavior such as the medial preoptic area, the lateral septum, and the lateral habenula^{45,46} as measured by increases in c-Fos expression⁴⁷. Interestingly, the lateral habenula has recently been identified as a region that is sensitive to maternal stress⁴⁸ and specifically activated during pup distress calls⁴⁹."

Are the later behavioral deficits in pups in this study seen in Figure 4 due to the reduced maternal care and/or the psilocin exposure in the pup? Can these contributions be isolated and tested?

Change: Experiment

To understand whether the later behavioral deficits in mice were due to psilocin exposure in the pup, we carried out an additional study whereby pups were directly injected with psilocin on postpartum day 7, the day they would have been exposed via their mother's breastmilk. Mice were then aged to adulthood and assessed in an identical battery to that used to identify long-term behavioral deficits in offspring of dams exposed to psilocybin. We found that this intervention perfectly mimicked the effects of maternal psilocybin exposure on individual tasks (i.e. produced anhedonia in the sucrose preference test, but no other effects). It did not, however, recapitulate the overall behavioral deficits induced by maternal psilocybin exposure, suggesting that perhaps both direct exposure and maternal care impairments are necessary to produce a broader behavioral deficit phenotype. We have added this study to Figure 5 of the manuscript and added relevant discussion.

Reviewer #3 (Remarks to the Author):

The manuscript by Hatzipantelis et al describes a series of experiments in mice that aimed to determine whether psilocybin was effective at improving the disruptions to maternal care behaviours elicited by stress, as well as evaluating effects on anxiety-like behaviours in dams and their adult offspring. The authors employed standard batteries of behavioural test to determine these outcomes, in combination with a novel postpartum stress paradigm that had been previously validated by this group to interfere with maternal behaviour. These tests confirmed that stress exposure impaired maternal care behaviours, but that psilocybin administration did not prevent impairments associated with social stress. In fact, psilocybin administration in mouse dams during the postpartum period had long lasting adverse effects on mood related behaviours in both mothers and adult offspring.

These results represent an important new avenue of research into psychedelics as therapies, suggesting a potential risk of adverse consequences for people with peripartum mental illness. The candid report of these (possible) negative implications is particularly commendable in the current climate of extreme hype about the potential benefits of psychedelic substances across mental health diagnoses, without much attention paid to possible risks. The fact that this patient population (i.e. women with postpartum depression) are being actively recruited for clinical trials using psilocybin analogues makes this work especially timely.

The methodology, analysis and interpretation of results are sound. However, a major criticism of this manuscript is that it does not include an assessment of potential mechanisms that might be responsible for why post-partum female responses to psilocybin with increased anxiety-like behaviour, in contrast to its anxiolytic effects in virgin females. It seems plausible that the biological/molecular mechanisms that completely reverses the responses of parous females to psilocybin may relate to the reduction in head twitch responses (shown in Supplementary Fig 1). This could reflect a change in the serotonin receptor binding profile of psilocybin in parous dams (i.e. directed away from the 5-HT_{2A}) that is critical to examine in the

context of a) altered sex hormone profiles compared to the virgin state and b) the potential that estrogen and serotonin receptor interactions mediate the different behavioural responses to psilocybin in parous female mice.

Change: Experiment

To understand whether the differences between virgin females and postpartum females were due to differential engagement of serotonin receptors, we carried out several additional experiments. First, we performed a pharmacodynamics experiment comparing the head-twitch response of postpartum and virgin females, which validated that postpartum females produce a weaker psilocybin-induced head twitch response, although this is still reliant on binding to 5-HT₂ receptors as it was equally abolished by ketanserin in both reproductive states. We have added this study as Figure 6 of the manuscript and added relevant discussion. We also performed an experiment that assessed mRNA expression in the cortices of both virgin females, males, and postpartum females, finding that postpartum females exhibited downregulation of mRNA encoding 5-HT 1A, 2A and 2C receptors as well as *bdnf* when compared to either virgin females or males. We have added this as Figure S5 in the manuscript and added relevant discussion throughout. Given that all 5-HT receptors were downregulated, it is unlikely that psilocybin's pharmacology is being directed away from 5-HT_{2A}R and toward another 5-HT_R (e.g., 5-HT_{1A}R), but it may direct it toward other brain regions. As such, we added the following to the discussion: "Alternatively, the downregulation of serotonin receptors in the cortex of postpartum mice may alter the circuit activity of psilocybin, directing it toward activity in other brain regions. Indeed, psilocybin exposure increases the activity of brain regions critically implicated in parenting behavior such as the medial preoptic area, the lateral septum, and the lateral habenula^{45,46} as measured by increases in c-Fos expression⁴⁷."

Furthermore, it is impossible to disentangle the effects of psilocybin (psilocin) exposure during breastfeeding on adult offspring behaviour from the effects of disrupted maternal care, which has important implications for risk evaluation. This could be examined with a cross-fostering experiment (psilocybin-treated dams foster saline-treated pups and vice versa).

Change: Experiment

Given that a cross-fostering experiment would take years to complete, we aimed to assess whether the later behavioral deficits in mice were due to psilocin exposure in the pup by carrying out a study whereby pups were directly injected with psilocin on postpartum day 7, the day they would have been exposed via their mother's breastmilk. Mice were then aged to adulthood and assessed in an identical battery to that used to identify long-term behavioral deficits in offspring of dams exposed to psilocybin. We found that this intervention perfectly mimicked the effects of maternal psilocybin exposure on individual tasks (i.e. produced anhedonia in the sucrose preference test, but no other effects). It did not, however, recapitulate the overall behavioral deficits induced by maternal psilocybin exposure, suggesting that perhaps both direct exposure and maternal care impairments are necessary to produce a broader behavioral deficit phenotype. We have added this study to Figure 5 of the manuscript and added relevant discussion.

I also wonder whether the authors have validated that exposure to the intruder mouse in the post-partum period elicits a physiological stress response (i.e. increased CORT)? For understanding the mechanisms that drive changes in maternal care behaviours, it is important to know whether stress hormone signalling, as opposed to pheromone release or ultrasonic vocalisations, underlies the behavioural effects of this model.

Change: Text

While we have not assessed this directly, pup separation and psychosocial stress are known to increase CORT (PMID: 35050513) and that stress can dramatically impact the behavioral consequences of psilocybin exposure. We have therefore added relevant discussion to this effect: "Another explanation for the differential effects of psilocybin in virgin and postpartum female mice is that the postpartum dams were exposed to daily, predictable stress in the form of pup separation and/or the psychosocial stress model, while the virgin females were not. Recent studies have demonstrated that while acute, transient corticosterone elevations due to stress (or direct exposure) are necessary for any anxiolytic effects of psilocybin, chronic (28-day) corticosterone exposure prevented psilocybin-induced anxiolysis⁴⁸. While the present study was carried out on a far shorter timeframe, psychosocial stress and pup separation are known to increase corticosterone⁴⁹. Future studies comparing virgin and postpartum females in comparable stress paradigms will help determine whether the differences in psilocybin responsivity observed herein are due to physiological differences across reproductive status, or differences in stress-response/-resolution, or both."

Finally, for the comparisons between postpartum and virgin females in Figure 5, all dams received saline first, then psilocybin the day after, and maternal care was evaluated within-animal as a change between saline and psilocybin. However, is it also possible that the differences observed on PPD 7 are related to the additional "stressor" of the pup separation (2 hours) and reunion on PPD 6, when all dams received saline? Counterbalancing mice to receive either saline or psilocybin first on PPD 6 and then the other treatment on PPD 7 would help to clarify this.

Change: Experiment

We repeated this experiment as requested using a between-subjects study design as opposed to a within-subjects design to balance any stressors across treatment groups. We incorporated this into the previously described 2x2 study design

comparing the effects of both psilocybin and ketanserin on acute maternal care behaviors (immediately following head twitch response assessment). This reproduced the original finding when assessed within-subjects that psilocybin significantly increased the maternal care behavioral risk score and was added to Figure 6 of the manuscript along with associated discussion.